# DNL343 is an investigational CNS penetrant eukaryotic initiation factor 2B activator that prevents and reverses the effects of neurodegeneration caused by the integrated stress response

Ernie Yulyaningsih*[†], Jung H Suh*, Melania Fanok[†], Roni Chau, Hilda Solanoy, Ryan Takahashi, Anna I Bakardjiev, Isabel Becerra, N Butch Benitez, Chi-Lu Chiu, Sonnet S Davis, William E Dowdle[§], Timothy Earr, Anthony A Estrada, Audrey Gill, Connie Ha, Patrick CG Haddick[††], Kirk R Henne, Martin Larhammar[#], Amy W-S Leung, Romeo Maciuca, Bahram Memarzadeh, Hoang N Nguyen, Alicia A Nugent[¶], Maksim Osipov, Yingqing Ran, Kevin Rebadulla, Elysia Roche, Thomas Sandmann, Jing Wang[¶], Joseph W Lewcock, Kimberly Scearce-Levie**, Lesley A Kane[‡], Pascal E Sanchez[‡§]

Denali Therapeutics, South San Francisco, United States

*For correspondence:
ernieyulya@gmail.com (EY);
Suh@dnli.com (JHS)

[†]These authors contributed equally to this work
[‡]These authors also contributed equally to this work

Present address: [§]Altos Lab, Redwood City, United States; [#]BioArctic, Stockholm, Sweden; [¶]Genentech, South San Francisco, United States; **Cajal Neuroscience, Seattle, United States

[††]Deceased

**Abstract** The integrated stress response (ISR) is a conserved pathway in eukaryotic cells that is activated in response to multiple sources of cellular stress. Although acute activation of this pathway restores cellular homeostasis, intense or prolonged ISR activation perturbs cell function and may contribute to neurodegeneration. DNL343 is an investigational CNS-penetrant small-molecule ISR inhibitor designed to activate the eukaryotic initiation factor 2B (eIF2B) and suppress aberrant ISR activation. DNL343 reduced CNS ISR activity and neurodegeneration in a dose-dependent manner in two established in vivo models – the optic nerve crush injury and an eIF2B loss of function (LOF) mutant – demonstrating neuroprotection in both and preventing motor dysfunction in the LOF mutant mouse. Treatment with DNL343 at a late stage of disease in the LOF model reversed elevation in plasma biomarkers of neuroinflammation and neurodegeneration and prevented premature mortality. Several proteins and metabolites that are dysregulated in the LOF mouse brains were normalized by DNL343 treatment, and this response is detectable in human biofluids. Several of these biomarkers show differential levels in CSF and plasma from patients with vanishing white matter disease (VWMD), a neurodegenerative disease that is driven by eIF2B LOF and chronic ISR activation, supporting their potential translational relevance. This study demonstrates that DNL343 is a brain-penetrant ISR inhibitor capable of attenuating neurodegeneration in mouse models and identifies several biomarker candidates that may be used to assess treatment responses in the clinic.

## eLife assessment

This study presents **solid** evidence to support the effectiveness of the novel eIF2B activator DNL343 in mitigating the integrated stress response (ISR) and reducing neurodegeneration associated with ISR activation in two mouse models. These **important** findings offer promise for the potential use of DNL343 in treating vanishing white matter disease (VWMD), a rare condition resulting from eIF2B loss of function, and in addressing other neurodegenerative disorders characterized by ISR

involvement. The study also identified potential VWMD biomarkers, which hold significance for assessing disease progression and evaluating treatment responses.

## Introduction

Eukaryotic initiation factor 2B (eIF2B) is a heterodecameric protein that plays an indispensable role in regulating translation of mRNA into protein. This process is initiated by the assembly of GTP-bound eIF2 and methionine-carrying initiator tRNA (Met-tRNAi) into a ternary complex (TC) which, along with other initiation factors, delivers Met-tRNAi to the AUG start codon. Upon recognition of the start codon, GTP is hydrolyzed, and the resulting eIF2-GDP dissociates from the complex (*Gebauer and Hentze, 2004*; *Sonenberg and Hinnebusch, 2009*). The continual success of translation initiation is dependent on replenishment of active GTP-bound eIF2 through the exchange of GDP for GTP by eIF2B, a dedicated guanidine exchange factor for eIF2 (*Bogorad et al., 2018*; *Hanson et al., 2022*).

eIF2B activity is dynamically modulated by the integrated stress response (ISR), an evolutionarily conserved signaling pathway that is activated in response to diverse cellular stresses, such as viral infection, amino acid deprivation, protein misfolding, and oxidative stress (*Pakos-Zebrucka et al., 2016*; *Costa-Mattioli and Walter, 2020*). ISR activation centers on the phosphorylation of eIF2B binding partner, eIF2, which led to structural changes that alter its interaction with eIF2B, turning it from a substrate into a competitive inhibitor of the guanine nucleotide exchange factor (GEF) activity of eIF2B (*Adomavicius et al., 2019*; *Gordiyenko et al., 2019*; *Kenner et al., 2019*; *Schoof et al., 2021*). As a result, the availability of eIF2-GTP and the subsequent TC become limited, leading to attenuation of global translation. This also gives rise to the paradoxical increase in translation of some stress response proteins (*Harding et al., 2000*). A common feature of these proteins are upstream open reading frames (uORFs) in the 5′ region that act as translational decoys to the main protein coding sequence (MCS). Under non-stress conditions these regulatory elements are translated instead of the MCS. However, during ISR activation, when TC and initiation cofactors are limited, translation of these uORFs are bypassed in place of the MCS due to leaky ribosomal scanning (*Wethmar et al., 2010*; *Young and Wek, 2016*; *Chen and Tarn, 2019*; *Costa-Mattioli and Walter, 2020*). One such protein is activating transcription factor 4 (ATF4) (*Vattem and Wek, 2004*), the primary effector of the ISR which regulates the expression of a wide range of adaptive genes aimed at restoring homeostasis or, in the event that the stress signal cannot be mitigated, triggering apoptosis (*Walter and Ron, 2011*).

Dysregulation of the ISR pathway has pathological consequences. In human and rodent models, mutations that lead to persistent activation of the ISR are associated with neurological disturbances (*Harding et al., 2009*; *Borck et al., 2012*; *Abdulkarim et al., 2015*; *Kernohan et al., 2015*; *Skopkova et al., 2017*) and those that affect the eIF2B subunits cause a rare leukodystrophy called vanishing white matter disease (VWMD) (*van der Knaap et al., 2006*). Additionally, evidence of constitutive pathway activation has been found in neurodegenerative conditions like amyotrophic lateral sclerosis (ALS) (*Ilieva et al., 2007*; *Ito et al., 2009*) and Alzheimer's disease (*Hoozemans et al., 2005*; *Unterberger et al., 2006*; *Hoozemans et al., 2009*; *Stutzbach et al., 2013*; *Storkebaum et al., 2023*), suggesting that dysregulation of the ISR may contribute to these disorders.

Pharmacological modulation of the ISR is a key area of therapeutic research and has been accomplished through a variety of approaches, including inhibition of upstream ISR kinases, inhibition of eIF2 phosphatases, and activation of eIF2B (*Hetz et al., 2019*). Inhibition of upstream kinases such as PERK has been shown to cause on-target toxicity, which limits their therapeutic potential (*Halliday et al., 2015*). In contrast, eIF2B activators have proven to be well tolerated in animal models while also improving memory and neurological deficits (*Sidrauski et al., 2013*; *Chou et al., 2017*; *Abbink et al., 2019*; *Wong et al., 2019*; *Krukowski et al., 2020*). The first published eIF2B activator, ISRIB, binds to and stabilizes the eIF2B complex in its decameric form and promotes the GEF activity of eIF2B regardless of the presence of phosphorylated eIF2 (*Tsai et al., 2018*; *Zyryanova et al., 2018*). ISRIB and its analog, 2BAct, were also neuroprotective in an eIF2B mutant mouse model of VWMD that displays ISR-mediated neurodegeneration (*Abbink et al., 2019*; *Wong et al., 2019*). Despite this, further development of ISRIB is hampered by its poor solubility and oral bioavailability (*Wang et al., 2018*; *Abbink et al., 2019*), and 2BAct due to its cardiovascular liability in dogs (*Wong et al., 2019*).

DNL343 is an investigational small-molecule eIF2B activator that has completed a Phase 1 clinical trial in healthy participants (NCT04268784) and is now undergoing clinical development for ALS (NCT05006352, NCT05842941). In immortalized cells and iPSC-derived neurons expressing ALS-linked mutations that are exposed to stressors, DNL343 inhibits the ISR and both prevents and reverses stress granule formation (*Sun et al., 2023*). The high potency, selectivity, and pharmacokinetic profile of DNL343 across preclinical species make it an ideal tool for studying ISR inhibition in vivo (*Craig et al., 2024*).

Here, we show that DNL343 reduces ISR activation and neurodegeneration brought on acutely by an optic nerve crush (ONC) injury or chronically by a germline eIF2B loss of function (LOF) mutation in mice. We also show that DNL343 treatment, initiated at a late stage of disease which more accurately replicates a clinical dosing paradigm, reversed these ISR and neurodegeneration biomarkers and prevented premature mortality in the eIF2B LOF mutant. Further, we identified CNS biomarkers of the ISR and neurodegeneration that are measurable in biofluids and showed data that support their translational relevance in samples from patients with VWMD and potentially other indications that are mediated by ISR activation.

## Results

### DNL343 is a novel brain-penetrant ISR inhibitor

To study the in vivo effects of our investigational small-molecule eIF2B activator, DNL343 (*Figure 1A*), we first demonstrated its suitability for acute and chronic dosing and evaluated brain exposure in wild-type mice (*Figure 1B*). Following a single dose of DNL343 at 62.5 mg/kg via oral gavage, unbound concentrations in the brain were similar to those in plasma (*Figure 1C*), supporting high CNS penetrance in mice (*Craig et al., 2024*). Chronic dosing via a non-invasive self-administration route was then examined by formulating DNL343 in mouse chow. After 5 weeks on medicated chow, dose-dependent increases in plasma and brain exposure of DNL343 were observed in mice (*Figure 1D and E*). Plasma DNL343 concentrations were similar following 2 or 29 days of dosing, indicating that steady-state conditions were rapidly achieved and sustained throughout the dosing duration (*Figure 1D*). Further, unbound concentrations of DNL343 in the brain were comparable to those in plasma for all doses at terminal timepoint (*Figure 1E*). No adverse event/s or effects on body weight (*Figure 1F*) were observed during the study, indicating that chronic treatment with DNL343 was well tolerated.

### DNL343 reduced ISR activation and neuronal loss in an optic nerve crush injury model in mice

To determine the impact of DNL343 on ISR-dependent neurodegeneration, we first utilized an ONC injury model. This model was selected based on the precedence for ISR activation and reproducible degeneration within a relatively short time period following injury (*Larhammar et al., 2017*). ONC was performed on the left optic nerve of wild-type mice, following which mice were given either vehicle or increasing doses of DNL343 once daily via oral gavage. (*Figure 2A*). The mice were euthanized at 2 or 14 days following the ONC procedure. Unbound concentrations of DNL343 measured in plasma or brain were similar, again confirming CNS penetration (*Figure 2B*). DNL343 exposure increased in a dose-proportionate manner and demonstrated low variability between animals (*Figure 2B*). We then measured the expression of ISR-dependent transcripts some of which are known to be altered in this model (*Larhammar et al., 2017*; *Abbink et al., 2019*; *Wong et al., 2019*) in the retina to confirm DNL343 activity in vivo. We found that many of these transcripts were robustly increased in the ipsilateral retina vs undamaged contralateral control retinas at 2 days and, to a lesser extent, 14 days after the ONC (*Figure 2C*, *Supplementary file 1 and 2*). DNL343 modulated the expression level of these ISR-dependent transcripts in a dose-dependent fashion (*Figure 2C*, *Supplementary file 1 and 2*) confirming its ability to inhibit ISR in vivo. In addition, the highest dose of DNL343 at 12 mg/kg reduced axonal degeneration (*Figure 2D*) and the two highest doses, 3 and 12 mg/kg, led to significant reduction in retinal ganglion cell (RGC) loss as measured by immunofluorescence analysis of neuronal markers (*Figure 2E and F*). Together, these data demonstrate pharmacodynamic and neuroprotective effects of DNL343 in an in vivo model of acute injury.

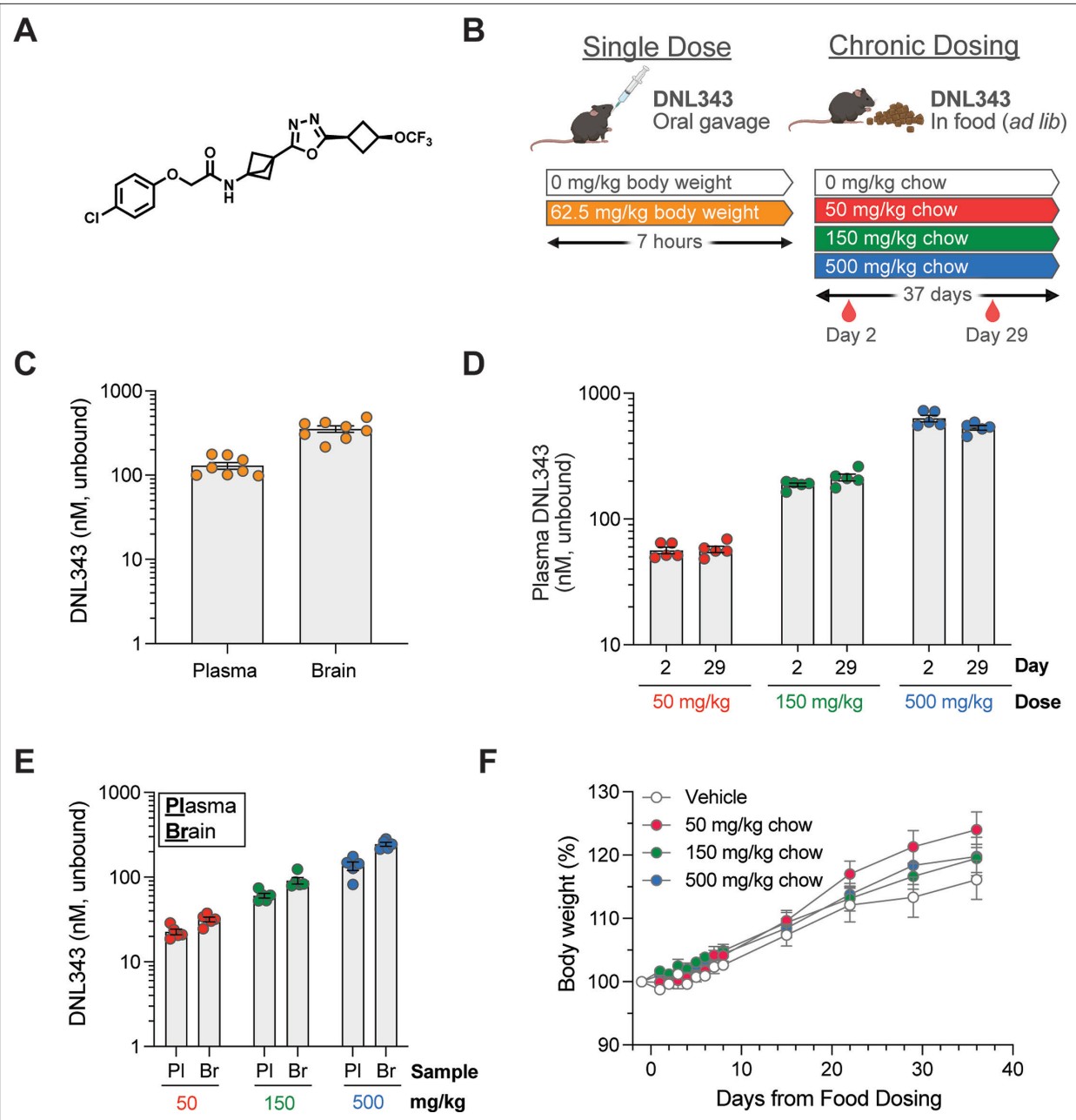

**Figure 1.** DNL343 is CNS penetrant and suitable for acute and chronic in vivo dosing. (**A**) Chemical structure of DNL343. (**B**) Schematic of DNL343 dosing studies in wild-type mice. Left: DNL343 (62.5 mg/kg) was administered via oral gavage and mice were taken down 7 hr after. Right: DNL343 was formulated in food at a concentration of 0, 50, 150, or 500 mg/kg chow. Plasma was sampled on days 2 and 29 of dosing at 1 hr after lights on (zeitgeber, ZT1) and animals were taken down on day 37 at ZT6. (**C**) DNL343 unbound concentration in the plasma and brain at 7 hr after a single administration of DNL343 at 62.5 mg/kg. (**D**) DNL343 unbound concentration in the plasma after 2 and 29 days of in-food dosing. (**E**) DNL343 unbound concentration in the plasma and brain after 37 days of in-food dosing. (**F**) Body weight of mice maintained on DNL343-formulated diet. Data are presented as mean ± SEM of N=8/group (**C**) or N=5/group (**D–F**).

## eIF2B LOF causes CNS ISR activation in the mouse

We next sought to develop a model that displays chronic ISR activation in the CNS to further test the effect of DNL343 treatment in vivo. To this end, we generated an eIF2B LOF mouse model by knocking in an *Eif2b5 R191H* point mutation that is homologous to the VWMD-causing human *R195H* mutation (*Fogli et al., 2002*). Homozygous (HOM) knock-in mice will hereafter be referred to as eIF2B HOM. Previous reports of independently generated *Eif2b5 R191H* mutant showed ISR overactivation

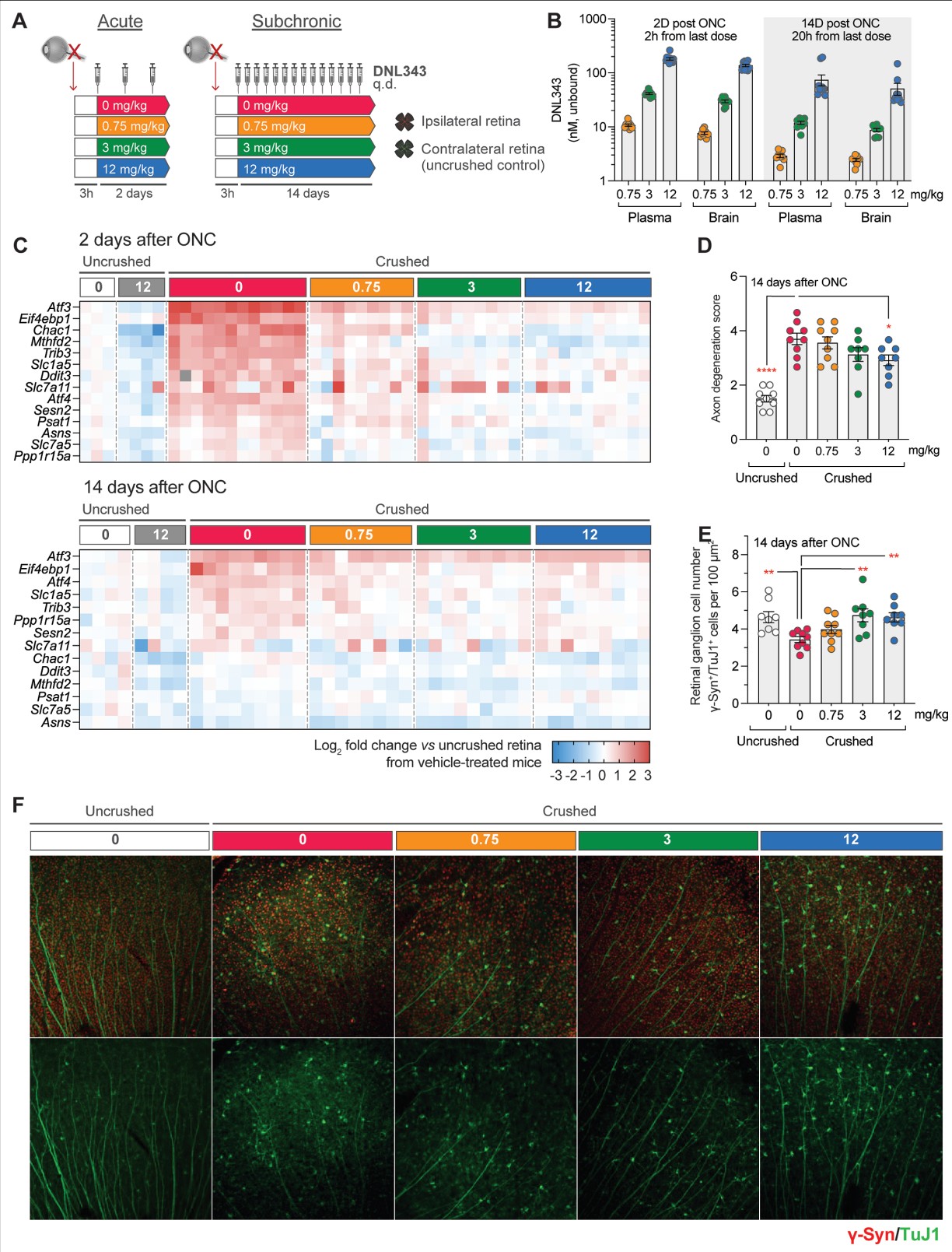

**Figure 2.** DNL343 engages the integrated stress response (ISR) and is neuroprotective in an acute ISR-driven neurodegeneration model, the optic nerve crush (ONC). (**A**) Schematic of the DNL343 dosing study in the ONC model. Wild-type mice underwent ONC procedure and received once daily DNL343 administrations via oral gavage 3 hr later. Animals were taken down at 2 or 14 days after the ONC procedure and retinas, ipsi- and contralateral to the crushed site, were collected. (**B**) DNL343 unbound concentration in the plasma and brain. (**C**) Heatmap showing expression level of

*Figure 2 continued*

ISR mRNA markers as assessed by multiplex qPCR in the retina, ipsilateral to the crush at 2 and 14 days after ONC, relative to control retinas from the uncrushed side. See **Supplementary files 1 and 2** for statistical data. (**D–E**) DNL343 reduced axonal degeneration score and retinal ganglion cell loss. (**F**) Representative immunofluorescent images of retinal ganglion cells as identified by TUJ1 (green) and γ-synuclein (red) immunoreactivity. Data are presented as mean ± SEM of N=8–12/group. Statistical significance for DNL343 effect was set at p<0.05 as determined by a one-way ANOVA followed by Dunnett's multiple comparison tests against crushed retina from vehicle-treated animals (*, p<0.05. **, p<0.01, ***, p<0.001, ****, p<0.0001).

The online version of this article includes the following source data and figure supplement(s) for figure 2:

**Figure supplement 1.** The integrated stress response (ISR) is activated in the brain of the eukaryotic initiation factor 2B (eIF2B) homozygous (HOM) mice.

**Figure supplement 1—source data 1.** Data associated with *Figure 2—figure supplement 1B*.

in the CNS and disease phenotypes relevant to human VWMD, such as demyelination and motor function deficit, that ultimately led to premature mortality (*Dooves et al., 2016*; *Abbink et al., 2019*; *Wong et al., 2019*). In line with this, characterization of select transcript markers in our eIF2B HOM model at 13–19 weeks of age confirmed ISR activation in the brain of mice HOM for the mutation, but not in heterozygous or wild-type littermates (*Figure 2—figure supplement 1A* and *Supplementary file 3*). The robust elevation of ISR markers were further confirmed in a bulk brain RNA sequencing analysis of a separate cohort of HOM mutant and wild-type littermates. Known ISR markers rank highly among the 492 protein-coding genes that showed significant differences between eIF2B HOM and wild-type animals (false discovery rate [FDR]<5%, *Figure 2—figure supplement 1B*). Genes reported as upregulated due to eIF2B LOF (*Wong et al., 2019*) were significantly enriched (gene set enrichment p-value<$10^{-20}$).

ISR activation appears to be most prominent in the CNS since tissues such as peripheral blood mononuclear cells or spleen have no marked upregulation of the ISR markers in HOM mice relative to wild-type controls (*Figure 2—figure supplement 1C and D* and *Supplementary file 4 and 5*). This is consistent with the reported CNS and peripheral phenotypes of independently generated models (*Dooves et al., 2016*; *Abbink et al., 2019*; *Wong et al., 2019*).

## Proteomic analysis reveals distinct brain and CSF signatures of ISR activation and neurodegeneration in eIF2B mutant mice

To further characterize the consequences of eIF2B LOF mutation in the mouse brain, we performed bulk brain proteomic characterization on an independent cohort of 11- to 15-week-old mice. Out of the 7939 proteins detected, we found 133 proteins that showed significant alteration in the eIF2B HOM genotype relative to wild-type controls (*Figure 3A* and *Figure 3—source data 1*). Of these, 45 proteins were also dysregulated on a transcriptional level (*Figure 2—figure supplement 1B*), suggesting some overlap between the transcriptomic and proteomic changes. In keeping with this, proteins that are regulated by ATF4, a transcription factor the levels of which are known to be ISR-dependent (*Pakos-Zebrucka et al., 2016*), are among those that are most significantly elevated, such as solute carrier (SLC) membrane transport proteins (SLC3A2, SLC7A1, SLC7A1), cystathionine γ-lyase (CTH), and phosphoserine aminotransferase 1 (PSAT1) (*Figure 3A*). Results from the proteomic analysis also show that disturbances in metabolic pathways appear to be the main consequences of the eIF2B LOF mutation on the brain proteome. Indeed, pathways related to amino acid transport and metabolism were found to be significantly (adjusted p<0.05) increased based on gene set enrichment analysis (GSEA) (*Figure 3C*). Taken together, these results show that eIF2B LOF leads to gross cerebral metabolic derangement that may contribute to the neurodegenerative phenotype associated with this model.

To understand how the ISR-dependent changes in brain proteins correlate to more clinically translatable fluid biomarkers, we also interrogated the CSF proteome of these eIF2B HOM mice. CSF tandem mass tag (TMT) proteomic analysis identified a total of 572 unique proteins, 472 of which were detected in both CSF and the brain. Univariate analysis identified a total of 44 proteins that were changed with unadjusted p<0.05 (*Figure 3B* and *Figure 3—source data 1*). Notably, the protein with the largest fold decrease in the eIF2B HOM CSF was MUP17 (log$_2$ fold change [FC]: –1.3, adjusted p<0.05), a major urinary protein (MUP) and a member of the broader lipocalin protein family, which has been shown to be decreased by ER stress (*Weber et al., 2022*). Several proteins associated with

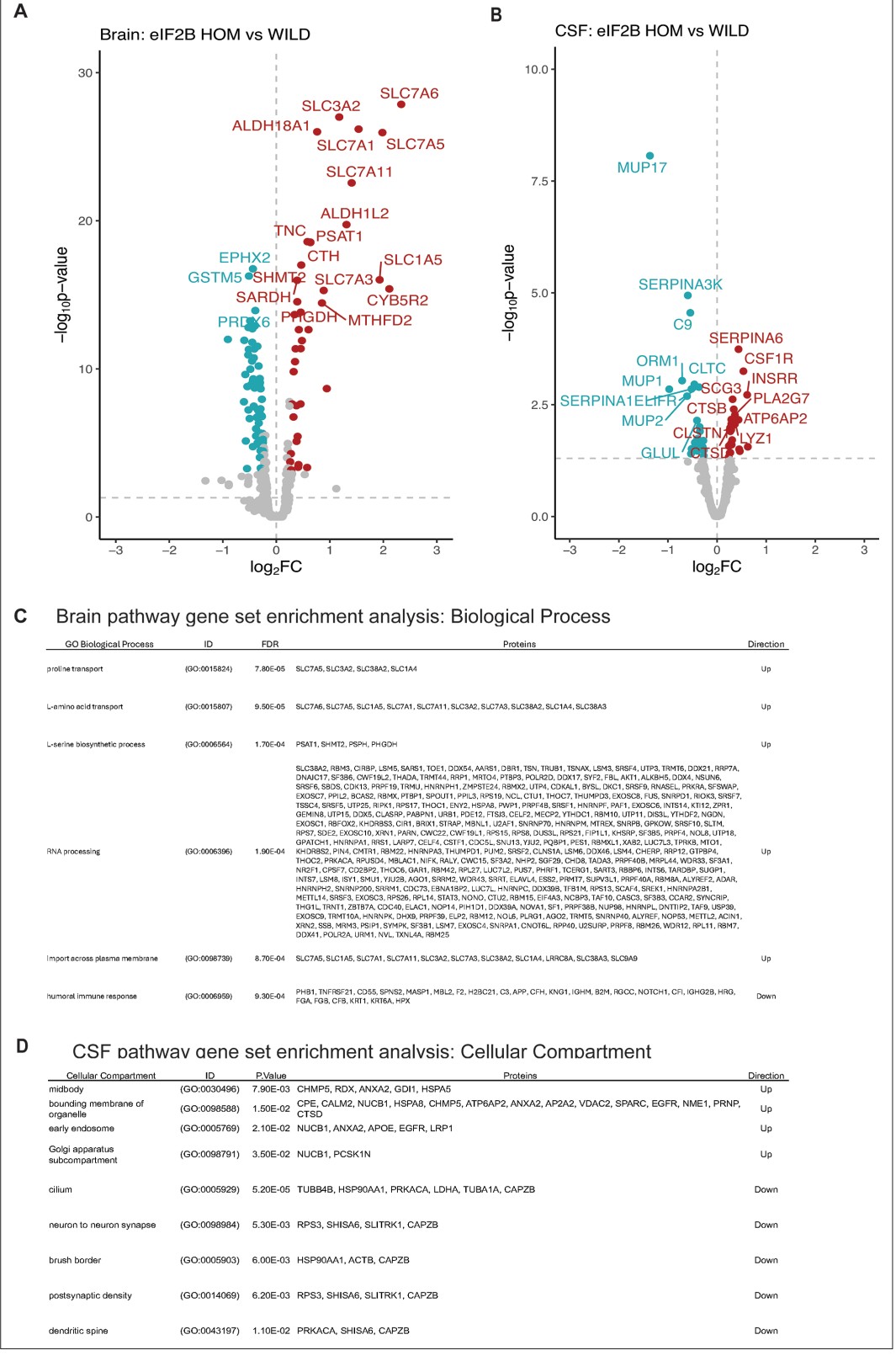

**Figure 3.** Bulk brain and CSF proteomics reveal distinct signatures of CNS integrated stress response (ISR) activation and neurodegeneration in the eukaryotic initiation factor 2B (eIF2B) homozygous (HOM) mice. (**A**) Volcano plot showing protein expression changes in the brain of 11- to 15-week-old eIF2B HOM mice relative to wild-type control with absolute $\log_2$ fold change (FC)>0.5 and p<0.00005. Blue and red circles denote either

*Figure 3 continued on next page*

*Figure 3 continued*

down- or upregulated proteins, respectively. Larger symbols denote top 20 altered proteins. In this analysis, 7939 proteins were detected. (**B**) Volcano plot showing protein expression changes in the CSF of 11- to 15-week-old eIF2B HOM mice relative to wild-type. Blue and red circles denote either down- or upregulated proteins, respectively. Top 20 significantly dysregulated proteins are labeled. 572 proteins were detected in this analysis. (**C**) Gene set enrichment analysis (GSEA) was performed to identify and detect specific GO biological processes that were altered in eIF2B HOM vs wild-type control in the brain. False discovery rate (FDR) shows adjusted p-value calculated based on Benjamini-Yekutieli (BY) procedure. (**D**) Enriched cellular compartment associated with dysregulated CSF proteome in eIF2B HOM mice. Unadjusted p-values are shown. Statistical significance for univariate analysis was set at p<0.05 as determined by robust linear model using the limma r package. N=7/group. For detailed statistical data from (**A**) and (**B**), see *Figure 3—source data 1*.

The online version of this article includes the following source data for figure 3:

**Source data 1.** Data relating to *Figure 3A and B*.

---

microglial activation (PLA2G7, CSF1R, LYSZ, and SIRPA) and intracellular organelle-associated proteins (CPE, CTSB, CLSTN1, CTSD, HEXA, GM2A, and ALDH2) were also higher in HOM mice relative to wild-type controls (*Figure 3B* and *Figure 3—source data 1*). Analysis of cellular compartment of origin of CSF proteins shows that proteins increased in the CSF were enriched for proteins associated with early endosome, Golgi, and membrane-bound organelle (*Figure 3D*). Results also showed decreases in CSF proteins associated with neuron-to-neuron synapse, postsynaptic density, and dendritic spine that may be reflecting accelerated neurodegeneration apparent in this model (*Figure 3D* and *Figure 3—source data 1*). Alterations in CSF levels of proteins associated with microglial activation and organelle proteins may indicate elevations in inflammatory burden associated with demyelination and neurodegeneration phenotypes seen in humans and other mouse models with mutations in the eIF2B protein (*Bugiani et al., 2018*; *Abbink et al., 2019*; *Wong et al., 2019*). Taken together, our data show very different proteome changes in the brain and CSF from eIF2B HOM mice and collectively, these data point to the CSF proteome representing biomarkers of demyelination and inflammation in the model while the changes in the brain appear to more directly represent cellular ISR dysregulation.

## DNL343 reduced brain ISR activation and prevented disease phenotypes in eIF2B HOM mice

Having characterized the eIF2B HOM model, we next tested the ability of DNL343 to inhibit ISR activation in these mice by initiating treatment at 10–17 weeks of age, before disease onset (*Figure 4A*) (based on clinical signs and plasma biomarker of neurodegeneration; *Figure 4F*). Animals received DNL343 medicated chow and intake was steady across all groups, achieving levels equivalent to oral gavage doses of 0, 0.3, 1, 3, or 10 mg/kg (*Figure 4—figure supplement 1A–C*). DNL343 exposure in the plasma showed a stable and dose-dependent profile from treatment weeks 2–12 (*Figure 4B*), with similar dose-matched exposure levels between wild-type and eIF2B HOM mice suggesting that DNL343 absorption was not altered in this model (*Figure 4B and C*). After 13 weeks of dosing, DNL343 exposure in the plasma and brain showed a dose-dependent increase (*Figure 4C*) that was accompanied by a dose-dependent reduction of ISR transcript markers in the brain of eIF2B HOM mice (*Figure 4D* and *Supplementary file 6*). These data further support the notion that DNL343 is effective at blocking chronic CNS ISR activation in vivo.

Consistent with the characterization of previously published eIF2B mutant models (*Abbink et al., 2019*; *Wong et al., 2019*), eIF2B HOM animals gain less weight over time compared to their wild-type counterparts. The two highest doses of DNL343 (3 and 10 mg/kg) significantly improved body weight increase (*Figure 4E* and *Figure 4—figure supplement 2A*), despite a modest reduction in food intake in these groups (*Figure 4—figure supplement 1B*). This improved disease phenotype was mirrored by a similar dose-dependent prevention of plasma NfL elevation, suggesting that DNL343 is neuroprotective in the model (*Figure 4F* and *Figure 4—figure supplement 2B*). Indeed, following 12 weeks of dosing, eIF2B HOM mice also showed a dose-dependent improvement in motor function as assessed by a balance beam test (*Figure 4G–I*). eIF2B HOM mice treated with DNL343 at doses of 3 and 10 mg/kg were indistinguishable from wild-type controls in the time taken to successfully cross the beam, the numbers of errors measured by foot slips, and falls from the elevated beam (*Figure 4G–I*).

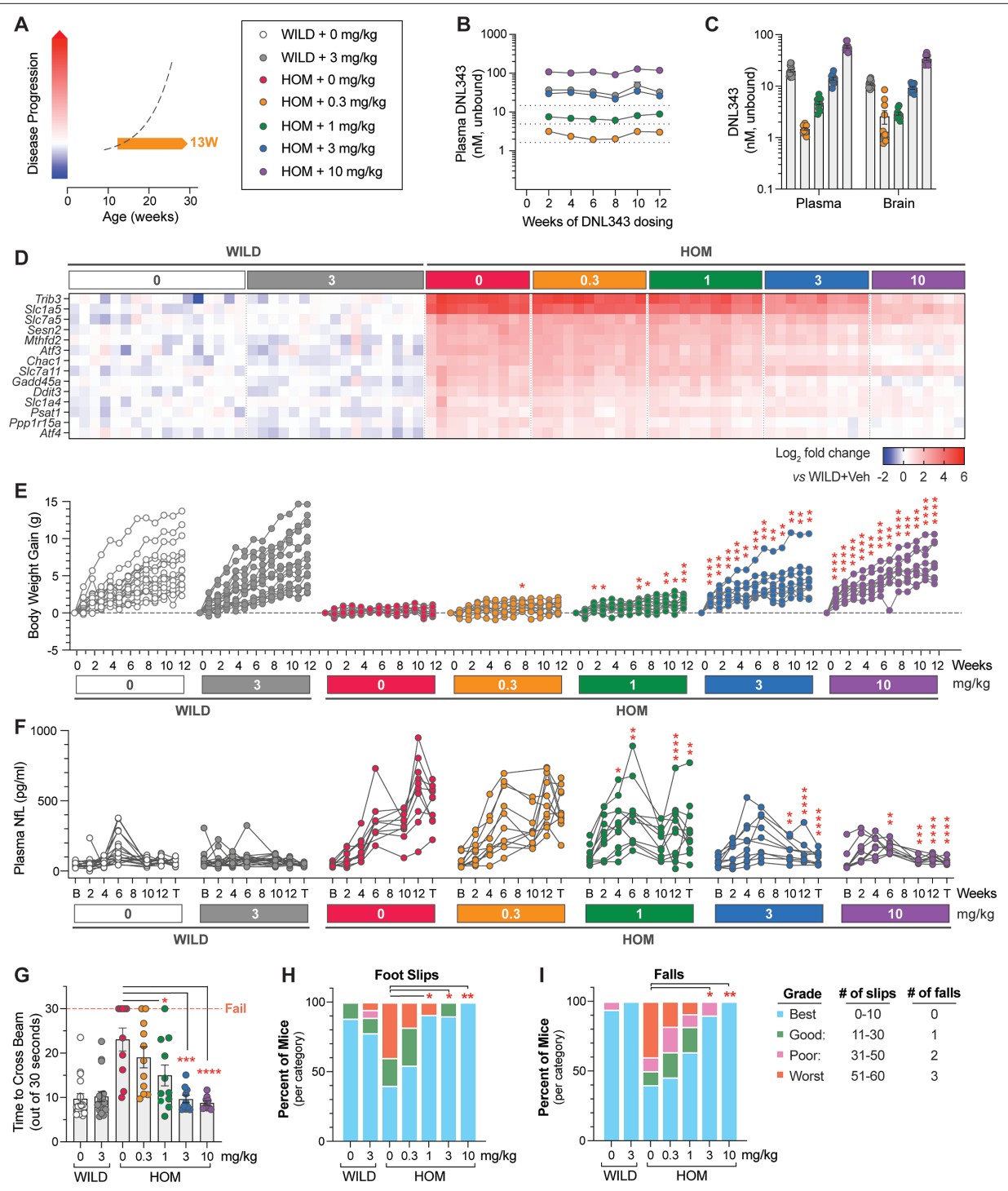

**Figure 4.** Chronic DNL343 dosing inhibits integrated stress response (ISR) overactivation in the brain, restores normal body weight, and prevents functional decline and neurodegeneration in the eukaryotic initiation factor 2B (eIF2B) mouse. (**A**) Schematic of the prophylactic DNL343 dosing study in the eIF2B homozygous (HOM) mouse model. DNL343 treatment was initiated in pre-symptomatic animals at 10–17 weeks of age. Mice self-administered DNL343-formulated rodent chow to achieve levels equivalent to oral gavage doses of 0, 0.3, 1, 3, or 10 mg/kg. (**B**) Plasma DNL343 exposure measured biweekly from weeks 2 to 12 of dosing. (**C**) Terminal plasma and brain DNL343 exposure after 13 weeks of dosing. (**D**) Heatmap visualization of select brain ISR transcript markers measured by multiplex qPCR in vehicle- and DNL343-treated wild-type and eIF2B HOM mice relative to vehicle-treated wild-type controls. See **Supplementary file 6** for statistical data. (**E**) Body weight gain measured weekly. (**F**) Plasma NfL levels assessed at baseline and throughout the study duration. B and T indicate baseline and terminal timepoints, respectively. (**G**) The balance beam test was performed after 12 weeks of dosing and the time to cross the beam out of 30 s were recorded. (**H–I**) The number of hindfoot slips and falls during the balance beam test. Data

*Figure 4 continued on next page*

*Figure 4 continued*

are presented as mean ± SEM of N=9–18 mice per group. Statistical significance for DNL343 effect was set at p<0.05 as determined by a mixed-effect analysis followed by Dunnett's multiple comparison tests against vehicle-dosed animals of the same genotype at matching timepoints (**E–F**) or one-way ANOVA followed by Dunnett's multiple comparison tests against vehicle-dosed animals of the same genotype (**G–I**). (*, p<0.05. **, p<0.01, ***, p<0.001, ****, p<0.0001).

The online version of this article includes the following figure supplement(s) for figure 4:

**Figure supplement 1.** DNL343-formulated food is well tolerated in both wild-type and homozygous mutant mice and targeted DNL343 are achieved.

**Figure supplement 2.** Alternative visualization of the effect of chronic DNL343 dosing on body weight and plasma NfL.

## DNL343 normalized proteins and metabolites in the brain and CSF of eIF2B HOM mice

We next evaluated the effect of chronic DNL343 treatment on the proteomic and metabolomic alterations in eIF2B HOM brains and CSF. The baseline genotype-associated differences observed in the brain and CSF proteomes were largely replicated in this older cohort of animals (5–7 months of age, *Figure 5—figure supplement 1A and B*, *Figure 3A and B*). In addition, targeted metabolomic and lipidomic analyses in the brain revealed profound increases in levels of oxysterols, such as 7-ketocholesterol, and cholesterol esters (CE) in eIF2B HOM mice (*Figure 5—figure supplement 1C*). These lipids tend to be increased in conditions of rapid demyelination (*Nugent et al., 2020*), consistent with the implication of the proteomics data and the phenotypes observed in VWMD patients (*Schiffmann et al., 1994*; *Bugiani et al., 2018*).

A 13-week treatment with DNL343 with doses that ranged from 0.3 to 10 mg/kg showed significant (p<0.05) dose-dependent normalization of proteins that were downregulated in brains of eIF2B HOM mice (*Figure 5A*, top cluster and *Figure 5—source data 1*). These included APOE, SLC7A10, CD63, TST, and ETNPPL that are associated with cholesterol, exosome, amino acid transport, detoxification, and glycerophospholipid metabolism. Chronic treatment also lowered key ISR proteins that were significantly upregulated in eIF2B HOM brains, including proteins involved in serine/glycine/1-carbon metabolism (PSAT1, ALDH1L2, CTH) and alanine/glutamine/leucine transporters (SLC3A2, SLC7A11, SLC7A3) (*Figure 5A*, bottom cluster and *Figure 5—source data 1*). Further, we identified proteins of amino acid metabolism pathway to be most significantly enriched in DNL343- vs vehicle-treated eIF2B HOM mice (*Figure 5B* and *Figure 5—source data 1*). CSF proteomic analysis also showed a clear DNL343 dose-dependent correction of protein biomarkers of microglial activation such as CSF1R, HEXB, CTSS, LYZ2, and CTSB (*Figure 5B*). As predicted by these results, biological processes that were most impacted by DNL343 were normalization of sulfur amino acid transport and nucleolar large rRNA transcriptions processes (*Figure 5D* and *Figure 5—source data 1*). As increased abundance of these proteins in the CSF have been associated with higher microglia activation, their correction by DNL343 may reflect downregulation of inflammation observed in the mutant mice. Furthermore, significant increases in 7-ketocholesterol and CE accumulation observed in eIF2B HOM mice were also dose-dependently corrected by chronic DNL343 treatment (*Figure 5C* and *Figure 5—source data 1*). Together these omics studies revealed that DNL343 can robustly correct biomarkers of ISR activation in the brain and markers of gliosis and demyelination in the CSF of eIF2B HOM mice at doses ≥3 mg/kg.

## Potential translational biomarkers are elevated in both eIF2B HOM mice and VWMD patient biofluids

We next sought to interrogate whether the changes observed in markers of ISR and immune activation in eIF2B HOM mice had potential translational relevance to human disease. A select number of proteins were chosen for evaluation in plasma and CSF samples from patients with VWMD (*Supplementary file 7*), a neurodegenerative disease that is driven by eIF2B mutations and chronic ISR overactivation (*van der Knaap et al., 2022*), with gene expression and protein levels also being measured in samples from DNL343-treated eIF2B HOM mice. Our selection included growth differentiation factor 15 (GDF-15), an ISR-regulated cytokine and biomarker of clinical inflammation and metabolic distress (*Lockhart et al., 2020*; *McGrath et al., 2020*), glial fibrillary acidic protein (GFAP), a marker of reactive astrocytes, and an emerging blood biomarker associated with neurodegeneration (*Abdelhak et al., 2022*), tissue inhibitor of metalloprotease 1 (TIMP-1), an astrocyte-released cytokine that has

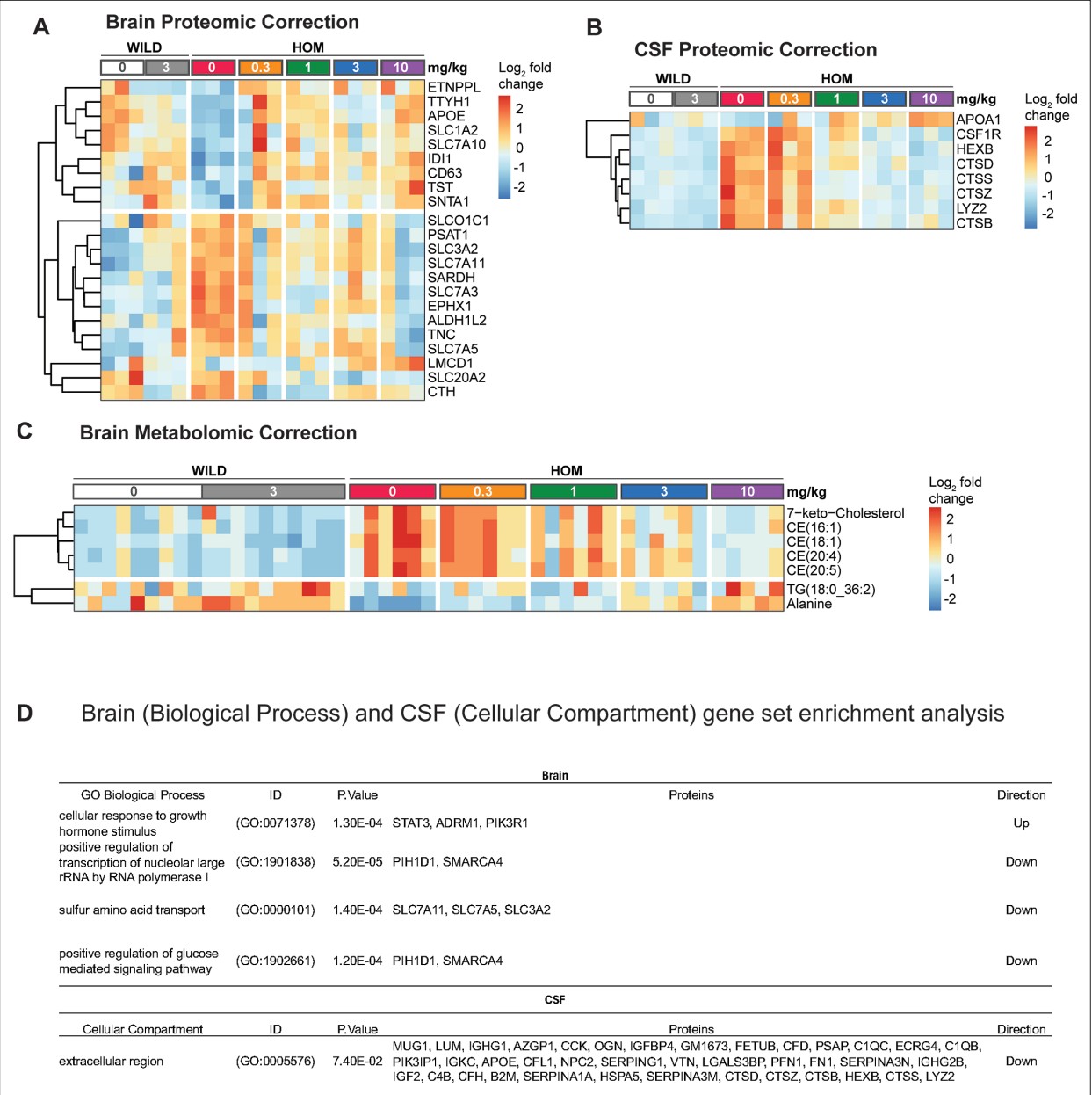

**Figure 5.** DNL343 dose-dependently normalized bulk brain and CSF protein and brain metabolite changes associated with integrated stress response (ISR) activation in the eukaryotic initiation factor 2B (eIF2B) homozygous (HOM) mouse. (**A**) Heatmap visualization of differential expression of proteins in eIF2B HOM mice vs wild-type controls following treatment with vehicle or DNL343 for 13 weeks (N=3/group). Wild-type mice received vehicle or DNL343 at the 3 mg/kg dose level. Features with significant (p<0.05) dose responses were selected and clustered based on Euclidean distances. Results show two main clusters of responsive protein that were up- (top) or downregulated (bottom) in a dose-dependent manner. (**B**) CSF proteomic shifts in the same mice shown in panel A are shown. Results show that significant elevation in abundance of CSF markers of microglial activation was decreased with DNL343 treatment at dose higher than 3 mg/kg body weight for 2 weeks. (**C**) Heatmap visualization of metabolomic changes in the bulk brain of eIF2B HOM relative to wild-type mice. Statistical significance was set at p<0.05 as determined by robust linear model. N=5–10/group. (**D**) Proteins detected in the brain and in CSF were ranked by logFC and the directionality of enrichment for specific biological processes in the brain and cellular compartment of origin of dysregulated CSF proteins. p-Values denote unadjusted p-values estimated from gene set enrichment analysis (GSEA). Protein column denotes detected pathway proteins corresponding to processes and compartment indicated. For detailed statistical data from (**A–C**), see *Figure 5—source data 1*.

The online version of this article includes the following source data and figure supplement(s) for figure 5:

**Source data 1.** Data relating to *Figure 5*.

*Figure 5 continued on next page*

*Figure 5 continued*

**Figure supplement 1.** Confirmation of integrated stress response (ISR)-associated proteomics changes in bulk brain and CSF in a cohort of 5- to 7-month-old mice, establishing evidence for metabolomic perturbations in the eukaryotic initiation factor 2B (eIF2B) homozygous (HOM) mice relative to the wild-type controls.

been implicated in neurodegenerative conditions (*Lorenzl et al., 2003*), and monocyte chemoattractant protein (MCP-1). These cytokines were largely not detected by TMT proteomics in the brain or CSF of the HOM mice as they are usually present in low abundance and therefore are challenging to detect in broad discovery proteomics method applied in this study. Here, we focused on antibody-based immunoassays that are better suited to specifically measure low abundant proteins.

Both mRNA and protein levels of GDF-15 were increased in the eIF2B HOM brains relative to wild-type controls and were dose-dependently reduced by DNL343 treatment (*Figure 6A and B*). These changes were replicated in a separate cohort of animals (*Figure 6—figure supplement 1A and B*) and were also detected in the mouse CSF (*Figure 6—figure supplement 1C*), suggesting that detection of this biomarker in the fluid could be a potential surrogate to assess CNS ISR activation and modulation. GDF-15 protein levels were also elevated in the CSF and plasma of VWMD patients relative to healthy controls (*Figure 6C and D*) even accounting for the age-dependent increase in this biomarker (*Lockhart et al., 2020*) that is particularly apparent in the plasma of patients for which longitudinal samples were available (*Figure 6—figure supplement 2A*). The astrocyte activation marker GFAP was elevated in eIF2B HOM mice with levels of mRNA in the brain and protein in the plasma both being corrected by DNL343 treatment (*Figure 6E and F*). In line with these data in the preclinical disease model, GFAP protein concentrations were higher in the CSF and plasma from VWMD patients relative to controls (*Figure 6G and H*). Although both TIMP-1 and MCP-1 levels were elevated in the eIF2B HOM mice, TIMP-1 levels were not altered in VWMD patients relative to controls in either the CSF or plasma and MCP-1 protein was only elevated in VWMD patient plasma (*Figure 6I–P*). Additionally, levels of other potential neurodegeneration biomarkers, interleukin 18 (IL-18), and soluble triggering receptor expressed on myeloid cells 2 (sTREM2), did not diverge in VWMD patients relative to controls (*Figure 6—figure supplement 2B and C*). On the other hand, NfL, a marker of neurodegeneration (*Gaetani et al., 2019*), was significantly elevated in the plasma and CSF of VWMD patients (*Figure 6S–R* and *Figure 6—figure supplement 2D and E*) mirroring the changes seen in the eIF2B HOM mice (*Figure 4F*). In sum, GDF-15, GFAP, and NfL demonstrated potential translational relevance by having increased gene expression or protein levels in eIF2B HOM mice relative to wild-type controls, which is reduced by administration of DNL343, as well as showing elevation in VWMD patient biofluids.

## DNL343 treatment initiated at advanced stage of disease rescued premature mortality and reversed biomarkers of neurodegeneration in the eIF2B HOM mice

Having shown that DNL343 can robustly prevent ISR activation and disease phenotypes in the eIF2B HOM mouse, we tested the effectiveness of DNL343 to reverse disease in older eIF2B HOM mice after disease onset (based on clinical signs and plasma NfL; *Figure 4*). DNL343 was administered in food at a concentration of 100 mg/kg chow for 4 weeks starting when the animals were 19–26 weeks of age, which corresponded to an advanced disease stage (*Figure 7A*). As expected, and previously established, this led to robust unbound plasma and brain exposure that is equivalent to an oral gavage dose of 10 mg/kg (*Figure 7B*, *Figure 4—figure supplement 1A*). Within only 4 weeks of treatment, DNL343 fully restored normal body weight gain (*Figure 7C*) and reduced the elevated expression of ISR transcripts in the eIF2B HOM brain (*Figure 7D* and *Supplementary file 8*), suggesting that administration of DNL343 can not only prevent ISR activation (*Figure 4D*), but also reverse it in animals with advanced disease. With this short duration of DNL343 treatment, neither motor function nor plasma NfL were fully rescued (*Figure 7—figure supplement 1A and B*). Given the robust effects on both ISR markers and weight gain, we evaluated the effects of longer treatment with DNL343 in this treatment paradigm starting when the animals were 24–32 weeks of age (*Figure 7A*, *Figure 7—figure supplement 2A*), when independently generated eIF2B mutant mice were approaching their survival limit (*Dooves et al., 2016*). In line with our hypothesis, 20 weeks of DNL343 treatment normalized the elevated plasma NfL level in eIF2B HOM mice (*Figure 7E*). This dosing paradigm also reversed

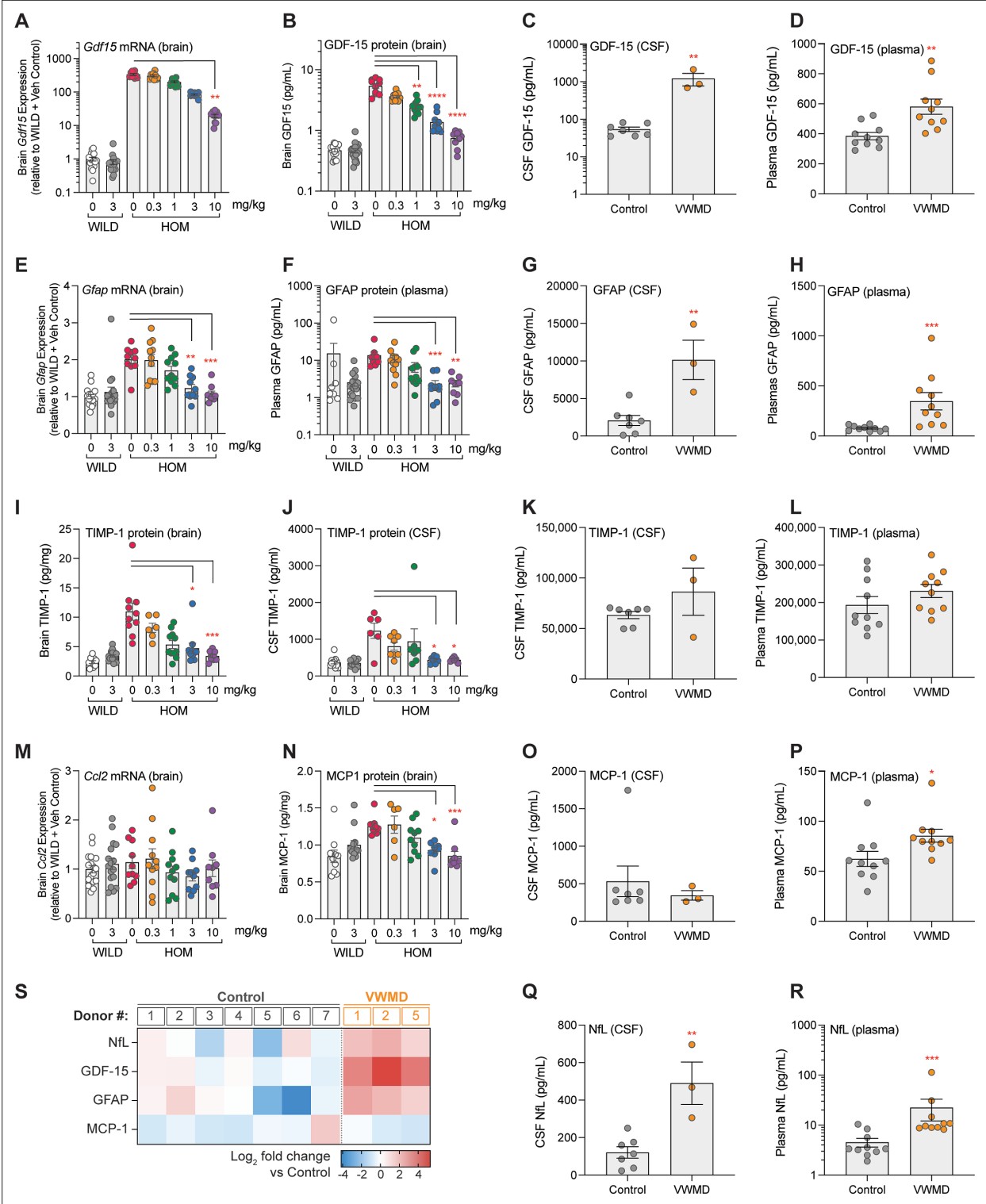

**Figure 6.** Exploratory biomarkers responsive to DNL343 in eukaryotic initiation factor 2B (eIF2B) mouse model and assessed in vanishing white matter disease (VWMD) in patient CSF and plasma. (**A–B**) Expression of *Gdf15* mRNA and GDF-15 protein in the brain of eIF2B homozygous (HOM) or wild-type mice treated with DNL343 or vehicle. (**C–D**) GDF-15 protein levels in the CSF and plasma of VWMD patients and healthy controls. (**E–F**) Expression of brain *Gfap* mRNA and plasma GFAP protein in eIF2B HOM or wild-type mice treated with DNL343 or vehicle. (**G–H**) GFAP protein levels in the CSF and plasma of VWMD patients and healthy controls. (**I–J**) TIMP-1 protein levels in the brain and CSF of eIF2B HOM or wild-type mice treated with DNL343 or vehicle. (**K–L**) TIMP-1 protein levels in the CSF and plasma of VWMD patients and healthy controls. (**M–N**) Expression of the *Ccl2* transcript, which encodes the MCP-1 protein, and levels of MCP-1 protein in the brain of eIF2B HOM or wild-type mice treated with DNL343 or vehicle.

*Figure 6 continued on next page*

*Figure 6 continued*

(**O–P**) MCP-1 protein levels in the CSF and plasma of VWMD patients and healthy controls. (**Q–R**) NfL protein levels in the CSF and plasma of VWMD patients and healthy controls. (**S**) Heatmap visualization of relative changes in NfL, GDF-15, GFAP, and MCP-1 in the CSF of VWMD patients vs healthy controls, presented in $\log_2$ scale. VWMD patient ID#s correspond across *Figure 6—figure supplement 2* (CSF) and *Figure 6—figure supplement 2* (plasma). Statistical significance was set at $p<0.05$. For all animal model panels, DNL343 dose is indicated on the x-axis and data is presented as mean ± SEM of N=9–18 mice per group. Statistical significance for DNL343 effect in the mouse model was determined by a one-way ANOVA followed by Dunn's multiple comparison tests against vehicle-dosed animals of the same genotype (*, $p<0.05$. **, $p<0.01$, ***, $p<0.001$, ****, $p<0.0001$). Data from VWMD patients and healthy controls is presented as mean ± SEM. Statistical significance for the difference between samples from VWMD patients and healthy controls was assessed on $\log_2$ fold change data using Welch's t test (*, $p<0.05$. **, $p<0.01$, ***, $p<0.001$, ****, $p<0.0001$).

The online version of this article includes the following figure supplement(s) for figure 6:

**Figure supplement 1.** Elevation in GDF-15 mRNA and protein levels in the eukaryotic initiation factor 2B (eIF2B) homozygous (HOM) brain and correction by DNL343 is reflected by changes in the CSF.

**Figure supplement 2.** Evaluation of biomarkers for vanishing white matter disease (VWMD) in patient CSF and plasma.

the increased level of plasma GFAP in eIF2B HOM mice and normalized it to wild-type control level (*Figure 7F*). We were unable to evaluate the impact of DNL343 on motor function by the balance beam test as animals of both genotypes were unable to complete the training after 8 weeks of treatment. This impairment appeared to be driven by distinct factors in the two genotypes: age-associated obesity in wild-type animals and severe motor impairment in the eIF2B HOM mice, irrespective of treatment. This and our earlier data (*Figure 4G–I*) suggest that DNL343 could prevent but not reverse functional deterioration, which is in line with our understanding of DNL343 mechanism of action that does not include neuronal regeneration, a therapeutic effect that is likely required for functional recuperation. Nonetheless, correction in the plasma biomarkers of neurodegeneration was accompanied by a robust extension of survival in the eIF2B HOM mice. As expected, in this 20-week study only 25% (N=3/12) of vehicle-treated eIF2B HOM mice survived to the end of the study. In contrast, 84.6% (N=11/13) of DNL343-treated eIF2B HOM mice survived beyond the study cessation (*Figure 7G* and *Figure 7—figure supplement 2B*), confirming that DNL343 can resolve ongoing neurodegeneration and neuroinflammation leading to prolonged lifespan in this model. Together these data demonstrate that DNL343 is a potent, CNS penetrant, ISR inhibitor that can correct or reverse downstream effects of chronic ISR overactivation.

## Discussion

Increasing evidence suggests that the ISR plays a causative role in a broad range of CNS disorders such as ALS and VWMD (*Costa-Mattioli and Walter, 2020*; *Storkebaum et al., 2023*). DNL343 is an investigational small-molecule eIF2B activator that is developed to address ISR-mediated neurodegeneration. Here, we showed that DNL343 robustly inhibited the ISR in the CNS whether activated acutely by mechanical crush injury to the optic nerve or chronically by an eIF2B LOF mutation. DNL343 also exerted a neuroprotective effect in both models by preventing either degeneration of the RGC or elevation in a plasma biomarker of neurodegeneration, NfL, and motor function deterioration. Initiating DNL343 treatment after disease onset in the eIF2B LOF model led to equally robust inhibition of the ISR and even reversed the elevated levels of plasma markers of neurodegeneration, NfL, and neuroinflammation, GFAP, and prolonged survival in the model. Our study also put forth candidate biomarkers that were identified in the mouse model and cross-validated in clinical samples from patients with VWMD to potentially enable clinical assessment of CNS ISR activation and modulation, and their downstream consequences. Together these data demonstrate the therapeutic effects of DNL343 in nonclinical in vivo settings and present clinical biomarker candidates.

DNL343 is a novel small-molecule eIF2B activator that was engineered for high CNS penetration. Consistent with this, data across multiple dosing studies in mice showed that DNL343 achieved comparable unbound exposures in the plasma and brain. DNL343 exposure also showed a favorable dose-dependent profile with no notable differences between healthy animals and disease models. In a chronic setting, DNL343 steady-state concentration was rapidly achieved and maintained, and the compound was well tolerated at above neuroprotective exposure levels in the ONC and eIF2B LOF models. In line with its preclinical pharmacokinetic data, DNL343 showed low pharmacokinetic

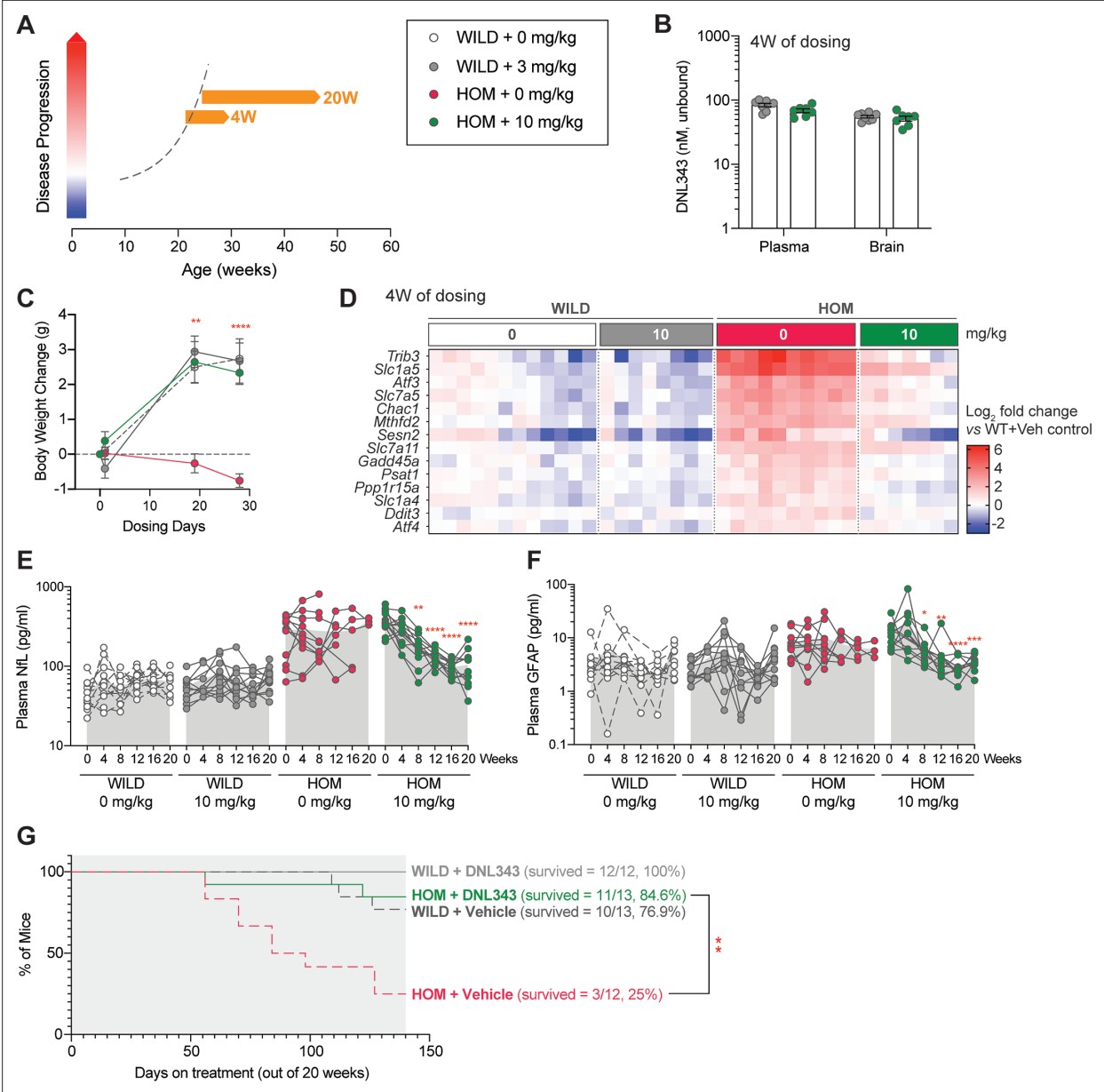

**Figure 7.** Reduction in integrated stress response (ISR) activation in the CNS is associated with reversal and normalization of plasma biomarker of neurodegeneration and extension of life when DNL343 treatment was initiated toward end of disease stage in the eukaryotic initiation factor 2B (eIF2B) homozygous (HOM) mouse. (**A**) Schematic of DNL343 therapeutic dosing studies in the eIF2B HOM mouse. DNL343 treatment was initiated when animals already presented with disease phenotypes at 19–26 or 24–33 weeks of age for 4 or up to 20 weeks, respectively. Mice self-administered DNL343-formulated rodent chow to achieve levels equivalent to oral gavage dose of 10 mg/kg. (**B**) DNL343 exposure was assessed in terminal plasma and in the brain after 4 weeks of dosing. (**C**). Body weight was measured at baseline and up to 4 weeks of dosing. (**D**) Heatmap visualization of select ISR transcript markers in the brain following vehicle and DNL343 treatment for 4 weeks. See ***Supplementary file 8*** for statistical data. (**E–F**) Plasma biomarker of neurodegeneration, NfL, and neuroinflammation, GFAP, was measured every 4 weeks for up to 20 weeks of dosing. (**G**) Kaplan-Meier graph shows that the reduced lifespan of the eIF2B HOM mice was extended when DNL343 treatment was initiated at 24–33 weeks of age. Data are presented as mean ± SEM of N=7–10 mice (panels **B–D**) or 10–13 mice per group (panels **E–G**). Statistical significance for DNL343 effect was set at $p<0.05$ as determined by a mixed-effect analysis followed by Šídák's multiple comparison tests against vehicle-dosed animals of the same genotype at matching timepoints (**C**) or against baseline values for each group (**E–F**). Effect of DNL343 on eIF2B HOM survival was assessed with log-rank test (*, $p<0.05$. **, $p<0.01$, ***, $p<0.001$, ****, $p<0.0001$).

The online version of this article includes the following figure supplement(s) for figure 7:

**Figure supplement 1.** Four weeks of DNL343 treatment initiated at late stage of disease did not rescue motor function but led to a trend of plasma NfL reduction.

*Figure 7 continued on next page*

Figure 7 continued

**Figure supplement 2.** Baseline characteristic and age at the end of the 20-week DNL343 therapeutic dosing study in the eukaryotic initiation factor 2B (eIF2B) homozygous (HOM) model.

variability and high CSF distribution in healthy volunteers and participants with ALS, supporting its progression as an investigational therapeutic for CNS indications (*Sun et al., 2023*).

Data from our in vivo models showed that DNL343 can inhibit the ISR in multiple mouse models. In an acute ONC model, we showed significant ISR upregulation in the retina 2 days following ONC as previously reported (*Larhammar et al., 2017*). This activation persisted but was less pronounced after 14 days suggesting that the ISR is activated most strongly as an early response to acute injury. At both timepoints, DNL343 dose-dependently normalized the ISR pathway leading to protection against RGC loss with mice treated at doses ≥3 mg/kg showing numbers of surviving RGCs that were comparable to that of uncrushed controls. The neuroprotective potential of ISR inhibition in the model was previously demonstrated using ISRIB (*Larhammar et al., 2017*). In contrast to our data however, treatment with ISRIB at a higher and more frequent dose of 10 mg/kg (b.i.d.) led only to a partial correction of *Chac*1 (an ISR marker that is assessed in both the published data and our study) and RGC survival. This points to improved potency as a potential differentiator between DNL343 and the previous generation ISR inhibitor. Another consideration could be that the greater extent of ISR activation seen in the published data (*Larhammar et al., 2017*) exceeded the cellular threshold beyond which ISRIB would have no effect. To date, such a phenomenon had only been demonstrated in vitro in the context of sodium arsenite-induced stress (*Rabouw et al., 2019*); whether it also underlies the discrepancy in these in vivo studies remains to be evaluated. Other possible explanations include greater unbound DNL343 exposure in the brain owing to greater CNS penetration and/or lower protein binding, and improved cellular potency (*Craig et al., 2024*).

The robust ISR inhibition and neuroprotection in the acute nerve injury model led us to study the effect of DNL343 in chronic ISR-dependent neurodegeneration in an eIF2B LOF model. Our independently generated eIF2B LOF mouse exhibits features that are consistent with previously reported models, namely prominent ISR activation in the CNS that was detectable in early adulthood and persisted throughout life, accompanied by stunted body weight growth that does not appear to be driven by hypophagia and motor function impairment (*Dooves et al., 2016*; *Abbink et al., 2019*; *Wong et al., 2019*). In addition, we showed a progressive escalation in the level of a plasma neurodegeneration biomarker, NfL, that corresponded to the degree of motor function loss. DNL343 treatment, initiated before overt disease manifestation, dose-dependently normalized these phenotypes with doses ≥3 mg/kg reverting body weight gain, plasma NfL level, and motor function to wild-type-like levels. While preventative modulation of some of the endpoints shown in this study is consistent with published literature utilizing other eIF2B activators (*Abbink et al., 2019*; *Wong et al., 2019*), our data provided further insight to DNL343 effects not previously explored. Our longitudinal data on plasma NfL levels throughout the dosing study allowed us to observe that DNL343, at doses of 3 and 10 mg/kg, not only led to reduction but reversal of plasma NfL elevation that hinted at the potential for this molecule to reverse the consequences of chronic ISR activation. Indeed, when dosed in a therapeutic paradigm starting at ages that corresponded to end of life (*Dooves et al., 2016*), DNL343 not only suppressed ISR activation in the eIF2B LOF brain but also robustly reversed the elevated levels of plasma NfL and GFAP. Reduction in these markers were apparent after 8 weeks, but not at the earlier timepoint of 4 weeks of treatment and this timeframe appears consistent between the prophylactic and the therapeutic dosing studies, suggesting that resolution of active neurodegeneration and neuroinflammation is predicated on correction of upstream consequences of CNS ISR activation. Further, 4 weeks of therapeutic dosing with DNL343 did not restore motor function. While this could not be further tested, we postulate that prolonging the treatment may not lead to further improvement as motor neurons likely were already degenerated at the time of treatment initiation. On the other hand, longer treatment with DNL343 in the clinically relevant therapeutic dosing paradigm extended the otherwise shortened lifespan of the eIF2B LOF mouse with most mice surviving past 50 weeks of age when the study was terminated. These key insights that DNL343 can robustly inhibit the ISR reverse disease biomarkers and extend lifespan in an in vivo model when treatment was administered after the onset of disease has direct clinical relevance and is a major step forward in understanding the therapeutic potential of eIF2B activators.

We conducted both targeted and unbiased analyses using brain, CSF, and plasma samples from DNL343-treated eIF2B LOF mice to discover biofluid-based biomarkers that could be harnessed to monitor CNS ISR activation and modulation, and their downstream consequences in the clinic. We found a disconnect between proteins that were altered in the eIF2B LOF mouse CSF relative to those in the brain. Multi-omics approach undertaken allowed us to cross-validate ATF4-regulated changes in expression of cellular amino acid transport and metabolism at transcriptomic, proteomics level leading to phenotypic alterations in cystathionine and alanine levels. While bulk brain transcriptomic and proteomic analyses show consistent changes in transcripts and proteins that belong to the ISR and amino acid metabolism pathways, as previously published (*Wong et al., 2019*), our CSF proteomic findings identify proteins that represent downstream disease consequences such as demyelination and neuroinflammation. These findings were replicated in two cohorts of animals at 3–4 and 5–7 months of age which correspond to stages before and when disease presentation becomes evident in the eIF2B LOF model, respectively, speaking to the robustness of the findings but also highlighting the early onset of neurodegeneration in the model that precedes functional manifestation. Chronic treatment with DNL343 corrected these genotype-associated changes in the respective samples. Targeted analyses of proteins led us to discover additional markers of the ISR and neuroinflammation that could be measured in biofluids. Increased transcript or protein level of GDF-15, GFAP, and TIMP-1 in the eIF2B LOF brain and their dose responsive modulation by DNL343 were reflected by their changes in the CSF (GDF-15 and TIMP-1) and plasma (GFAP). Of these, we showed that GDF-15, a transcriptional target of ATF4, and GFAP, a marker of neuroinflammation and reactive astrocytes, were also elevated in the CSF and plasma of patients with VWMD, a rare neurodegenerative indication that is caused by biallelic recessive mutations in *EIF2B1-5* and in which the ISR is a central mechanism. While astrocytes are heavily implicated in the pathophysiology of VWMD (*Dietrich et al., 2005*; *van der Knaap et al., 2006*), astrocyte dysfunction is also increasingly being recognized to contribute to the initiation and progression of diverse neurodegenerative disorders (*Brandebura et al., 2023*), so GFAP is of particular interest as a CNS biomarker. NfL, a marker of axonal degeneration that is increased and correlated with disease progression in other neurodegenerative conditions (*Huang et al., 2020*; *Uphaus et al., 2021*; *Chelban et al., 2022*) and in the eIF2B LOF mouse model, was also significantly elevated in both the plasma and CSF from VWMD patients. Inflammatory markers, such as TIMP-1 and MCP-1, that we identified in the mouse model based on changes relative to wild-type control and dose responsiveness to DNL343 were unchanged in samples from patients with VWMD, highlighting contrasts that may be driven by species-specific differences. Nonetheless, the eIF2B LOF mouse serves as an important tool that models the biological consequences of chronic ISR activation in the CNS and from which biomarkers identified have broad applicability in other neurodegenerative indications. To this point, our DNL343 studies in two in vivo models allowed us to observe commonality in ISR transcript markers despite the distinct modes of activation and CNS tissues (acute activation in the retina after ONC and chronic activation in the eIF2B HOM brain). To our knowledge, our study is the first to perform multi-omics characterization of the CNS-adjacent and clinically accessible CSF in an ISR-dependent model of neurodegeneration. The biofluid-based candidate biomarkers we cross-identified in patients with VWMD – GDF-15, GFAP, and NfL – could serve as potential biomarkers to monitor the extent and consequences of ISR activation and therapeutic modulation in the clinic.

## Materials and methods

CSF and plasma from VWMD patients and healthy controls: CSF and plasma from VWMD patients were obtained under a material transfer agreement with Baylor Scott & White Research Institute (Dallas, TX, USA). Samples were collected 15–25 years prior to analysis. Plasma EDTA from healthy controls obtained from BioIVT, Allcells, and Denali Phase 1 clinical trial: NCT04268784. CSF from healthy controls obtained from Biochemed and Denali repository. Healthy control samples were collected 0–2 years prior to analysis.

### Quanterix Simoa assays in human samples

For CSF NfL measurements, CSF was diluted 100× and analyzed using Quanterix Simoa Neurofilament-Light (NfL) kit (Quanterix, # 103400) according to the manufacturer's instructions on the Quanterix SR-X instrument. For plasma measurements of NfL and GFAP, plasma was diluted 4× and analyzed

using Quanterix Simoa Neurology 4-Plex A (Quanterix, # 102153) kit according to the manufacturer's instructions on the Quanterix HD-X instrument. Measurements were interpolated against a calibration curve provided with the Quanterix assay kit produced using on-instrument software. All assays run with technical replicates and results averaged.

## Electrochemiluminescence-based detection (ECLIA) for the quantitation of (sTREM2) in human samples

MSD GOLD 96-well small spot streptavidin SECTOR plates (MSD, catalog # L45SA) were washed three times with 300 µl/well with wash buffer (1× TBST) using a plate washer. To each well, 1 µg/ml human TREM2 biotinylated antibody, polyclonal goat IgG (R+D, catalog # BAF1828) diluted in PBS was added and the plate was incubated at room temperature (RT) for 1 hr on an orbital shaker at 700 rpm. Plates were washed three times with 300 µl wash buffer and blocked with 150 µl per well of MSD Blocker A (MSD, catalog # R93AA) for 2 hr at RT on an orbital shaker at 700 rpm. Plates were washed three times with wash buffer and sample or standard were added diluted in 25% MSD Blocker A in TBST. hTREM2-His-tagged ECD protein standards were prepared at 4× dilutions starting at 50,000 pg/ml. Plates were incubated overnight (16–24 hr) at 4°C on an orbital shaker at 700 rpm. Plates were washed as above and sulfo-tagged human TREM2 antibody, polyclonal goat IgG (R+D, catalog # AF1828) detection antibody diluted 1:5000 in 25% MSD Blocker A in TBST was added to each well. Plates were incubated for 1 hr at RT on an orbital shaker at 700 rpm. Plates were washed again and 150 µl per well of MSD Read Buffer T (MSD, catalog # R92TC-3) diluted 1:1 in water was added to each well. Plates were read on a 1201 MSD SECTOR 600MM plate reader (Meso Scale Discovery, catalog # IC1AA-0) within 30 min of adding buffer and data was captured in Methodical Mind (Meso Scale Discovery, version Fall 2020 [1.0.38]). Pg/ml concentrations for each sample were determined by MSD analysis software Discovery Workbench (Meso Scale Discovery, version LSR_4_0_13) using a four-parameter logistic model. All assays run with technical replicates and results averaged.

## MSD assays for additional biomarkers in human samples

CSF and plasma samples were analyzed according to the manufacturer's protocol using the following MSD kits: R-PLEX Human GDF-15 Antibody Set (MSD, catalog # F21YD-3), R-PLEX Human TIMP-1 Antibody Set (MSD, catalog # F21YO-3), V-PLEX Human MCP-1 kit (MSD, catalog # K151NND-2), and U-PLEX Human IL-18 kit (MSD, catalog # K151VJK-2). For CSF only, samples were analyzed according to the manufacturer's protocol using R-PLEX Human GFAP Antibody Set (MSD, catalog # F211M-3). Plates were read on a 1201 MSD SECTOR 600MM plate reader (Meso Scale Discovery, catalog # IC1AA-0) within 30 min of adding buffer and data was captured in Methodical Mind (Meso Scale Discovery, version Fall 2020 [1.0.38]). Pg/ml concentrations for each sample were determined by MSD analysis software Discovery Workbench (Meso Scale Discovery, version LSR_4_0_13) using a four-parameter logistic model. All assays run with technical replicates and results averaged.

## Animal care

All procedures in animals were performed with adherence to ethical regulations and protocols approved by Denali Therapeutics Institutional Animal Care and Use Committee. Mice were housed under a 12 hr light/dark cycle (lights on at 7 AM, lights off at 7 PM) with ad libitum access to water and a standard chow (Labdiet, # 5LG4) unless otherwise indicated. Whenever possible, animals were group-housed of up to five sex-matched animals/cage.

## DNL343 preparation for oral gavage dosing

DNL343 was prepared as a suspension in a 0.5% methylcellulose solution. Briefly, DNL343 was placed in individual glass vials to which purified water (half of the final volume) was added. The mixture was gently mixed using a probe sonicator at 40% amplitude for 10 times, each lasting for 20 s on pulse sonication. The result was a suspension with dispersed particles that was uniform throughout the product. Equal volume of a 1% methylcellulose (2× final target concentration, in half of final target volume) solution in purified water was added to the above suspension and vortexed until uniform. Suspension was visually inspected to be uniformly distributed. Dosing solutions were prepared on the first dosing day then weekly thereafter and were maintained at 4°C, on a laboratory rotator, until use.

DNL343 suspension was remixed on dosing instances and administered via oral gavage at a volume of 10 ml/kg body weight.

### DNL343 preparation for in diet dosing

DNL343 was blended in bacon flavored grain-based rodent diet (PicoLab Mouse Diet 20, catalog # 5058, gross energy 4.63 kcal/g) and pelleted by Bio-Serv. For every target dose in mg of DNL343/ kg body weight per day, DNL343 was formulated based on an estimated daily food consumption of 0.1 kg of chow per kg body weight per day (e.g. 30 mg/kg chow for 3 mg/kg body weight). Bacon flavored base diet was used as placebo control. Coloring was added to differentiate between dosages and to blind experimenters to the treatments.

### DNL343 in diet dosing

Animals were group-housed whenever possible. Control or medicated food was weighed every week and refreshed every 4 weeks at minimum. Food intake was estimated by taking the difference in hopper weight, normalized for total body weight in each cage and presented per diem. DNL343 intake was estimated by adjusting the food intake for DNL343 concentration in the food. DNL343 exposure in the plasma, collected via submandibular bleed as an in-life procedure or via cardiac puncture at terminal timepoints, and brain was measured using LC/MS/MS.

### Tissue collection

For all tissue collection, animals were humanely anesthetized with 2.5% tribromoethanol. Whole blood was collected via cardiac puncture into EDTA-coated tubes and spun down at 12,700 rpm for 7 min at 4°C before collecting the top plasma layer for analysis. CSF, if collected, was then extracted via cisterna magna puncture, spun down at 12,700 rpm for 7 min at 4°C, following which the supernatant was collected and stored in a new tube for analysis. Animals were transcardially perfused with ice-cold PBS. Tissues were then collected, weighed, frozen on dry ice, and stored at 80°C for subsequent analysis.

### Optic nerve crush

ONC was performed as previously described by *Watkins et al., 2013*, on 12-week-old wild-type male C57Bl/6J mice obtained from the Jackson Laboratory (USA). Animals were deeply anesthetized with inhaled isoflourane. The left eye of each animal was subjected to optic nerve injury via an intraorbital approach. An incision was made on the superior conjunctiva of the left eye, and the optic nerve was exposed. The crush injury was inflicted for 10 s using a 45° angled fine tip forceps 1–3 mm from the eyeball. Post-operatively, a drop of 0.5% proparacaine hydrochloride ophthalmic solution was applied to the affected eye as an analgesic measure. A small amount of ophthalmic ointment was applied to the non-procedural eye prior to surgery to prevent drying. Unoperated right eyes (contralateral to the crushed site) were used as controls. Starting from 3 hr after the ONC procedure, mice were administered vehicle or DNL343 at doses of 0.75, 3, or 12 mg/kg once daily via oral gavage. Animals were humanely euthanized on day 2 after ONC (2 hr after the last dose) or on day 14 (20 hr after the last dose).

### Generation of the eIF2B LOF mouse

The *Eif2b5 R191H* knock-in mouse model was engineered and maintained on a C57BL/6J background by Ozgene (Australia) using the goGermline technology (*Koentgen et al., 2016*). Briefly, a targeting vector was designed against the mouse *Eif2b5* gene containing a CGC to CAC codon substitution to replace arginine for histidine at amino acid position 191 in exon 4. An FRT-flanked neomycin selection cassette was targeted between exons 5 and 6 and Flp-mediated recombination was used to excise this cassette. The colony was maintained by heterozygous breeding to obtain littermate wild-type and HOM knock-in animals. The mouse genotypes were confirmed by PCR.

### DNL343 preventative dosing in the eIF2B LOF model

10- to 17-week-old eIF2B HOM and their wild-type littermates were assigned to receive food containing DNL343 at concentrations of 30–100 mg/kg or base chow. Groups were balanced for differences in body weight, age, and sex within each genotype. Food weight and body weight were

assessed weekly. Plasma was collected at baseline and every 2 weeks starting at 2 hr after the onset of the light period. Motor function was assessed by a balance beam test after 12 weeks of treatment. Animals were humanely euthanized between 12.9 and 13.3 weeks of treatment. Mice from all groups are represented on each consecutive necropsy day.

## DNL343 therapeutic dosing in the eIF2B LOF model

19- to 26-week-old eIF2B HOM and their wild-type littermates were assigned to receive food containing DNL343 at a concentration of 100 mg/kg or base chow, accounting for differences in body weight, age, sex, and plasma NfL level within each genotype. The body weight was assessed at baseline and after 19 and 28 days of treatment. Motor function was assessed by a balance beam test after 27 days of treatment. Animals were humanely euthanized after 33 days of treatment.

Another cohort of eIF2B HOM and wild-type littermates were treated with DNL343-formulated diet (100 mg/kg chow) or base diet from 24 to 33 weeks of age for up to 20 weeks. Body weight was assessed at baseline and weekly after. Plasma was collected before and every 4 weeks after study initiation. Animals meeting any one of the following humane endpoints were euthanized: body weight loss of ≥10%, hindlimb paralysis, or inability to right itself within 20 s.

## Balance beam test

To assess balance and motor coordination, animals were trained to walk along a 98.5 cm long × 13 mm diameter wooden beam, elevated at 40 cm above the bench surface lined with a soft foam mattress (104×74×10 cm$^3$). A length of 80 cm was marked on the beam as the distance the animals were required to cross. A spotlight was placed at the starting end to serve as an aversive stimulus. A dark enclosure (black matte acrylic box, 20×15×20 cm$^3$) with bedding material and carton tube was placed at the opposite end to serve as an escape box with a more agreeable condition. A video camera was set in front of the starting end to record foot slips as the animal progressed across the beam.

Animals underwent three training runs 1 day prior to the testing. Each mouse was first placed in the escape box for about 1 min, then placed at an increasing distance away from the escape box over three trials. If an animal paused on the beam, or tried to turn and advance the wrong way, it was gently touched on the hindquarters and encouraged to keep moving toward the escape box. Animals were habituated to the experimental room for at least 1 hr prior to testing. To begin an experiment, each subject was placed on the starting end of the beam and allowed a cutoff time of 30 s to cross. The elapsed time was only recorded while the subject actively moved toward the finish line. If a fall occurred, the subject was restarted at the beginning of the beam. Each subject was given three attempts to complete the balance beam. If a subject failed all three attempts, it was assigned a time of 30 s and a value of three falls. The number of foot slips was quantified from the video recording only if the animal completed the task. The balance beam was cleaned with a 70% solution of isopropanol and dried between each subject. The escape box was cleaned with isopropanol and dried, and the bedding and carton tube replaced between animals of different cages. The experiment was video recorded. Time to traverse the beam and number of falls were recorded live by an experimenter who was blinded to the genotypes of the animals. Foot slip scoring was done post hoc by another experimenter, who was blinded to both genotypes and treatment, using the video recordings.

## DNL343 pharmacokinetics

Plasma and brain concentrations of DNL343 were measured using LC-MS/MS by Quintara Discovery. Brain samples were homogenized in three volumes of ice-cold water, then further diluted twofold with blank mouse plasma. An aliquot of 20 µl of plasma sample or plasma diluted brain homogenate was extracted with 100 µl of acetonitrile containing internal standard (warfarin). The mixture was vortexed on a shaker for 15 min and subsequently centrifuged at 4000 rpm for 15 min. An aliquot of 70 µl of the supernatant was mixed with 70 µl of 35% acetonitrile in water with 0.1% formic acid for injection to the LC-MS/MS. Calibration standards and quality control samples were prepared by dilutions of the 2 mg/ml DNL343 stock solution with 70% acetonitrile added to blank mouse plasma. These standards and QC samples were extracted alongside unknown samples in each bioanalytical batch. The final extracts were analyzed by reversed-phase liquid chromatography and negative electrospray ionizations (LC-MS/MS) with multiple reaction monitoring of the test article and the internal standard. The standard curve was fitted by linear regression to quantify DNL343 in the matrix using Analyst 1.6.3

software (AB Sciex). Total DNL343 concentrations were adjusted for brain or plasma protein binding (fraction unbound, $f_u$) that were determined in vitro via an ultracentrifugation method and expressed as unbound (free DNL343) exposures.

## qPCR gene expression analysis

For analysis of gene expression after ONC, retinas from the left and right eyes were collected into RNAprotect (QIAGEN, catalog # 76104) and stored at RT until further processing. RNA from the retina was isolated using the RNeasy Plus Micro kit (QIAGEN, catalog # 74034), and RNA from the brain using the RNeasy Plus Mini Kit (QIAGEN, catalog # 74134). RNA was reverse-transcribed into cDNA using the Fluidigm Reverse Transcriptase Master Mix (Fluidigm, catalog # 100-6298). Specific target amplification of cDNA was done per the manufacturer's recommendations as the initial step for the real-time qPCR on the Biomark HD system (Fluidigm, catalog # BMKHD-BMKHD). Briefly, 20X TaqMan assays were pooled and diluted with DNA Suspension Buffer (Teknova, catalog # T0221) to a final concentration of 0.1× of each assay. PCRs were carried out for each cDNA sample with the TaqMan assay pooled in 5 µl volumes, each containing 1.2 µl of the pooled assay mix, 1 µl PreAmp Master Mix (Fluidigm, # 100-5580), 1.5 µl of $H_2O$, and 1.25 µl of cDNA. The preamplification PCR was performed at one cycle of 95°C for 2 min, followed by 14 cycles of 95°C for 15 s and 60°C for 4 min, and then 4°C. Specific target amplification products from the PCRs were diluted 5× with DNA Suspension Buffer (Teknova, catalog # T0221). Gene expression analysis was carried out using the 96.96 dynamic arrays IFC (Fluidigm, catalog # BMK-M-96.96) on the Biomark HD system (Fluidigm, catalog # BMKHD-BMKHD). 10× assay mixture and sample pre-mix were prepared, with each assay being prepared in triplicates. Each 10× assay mix contained 2.5 µl of 20× TaqMan gene expression assay (Thermo Fisher) and 2.5 µl of 2× Assay Loading Reagent (Fluidigm, catalog # 100-7611). All qPCR TaqMan probes (FAM) were ordered from Thermo Fisher Scientific. Sample pre-mix was made by combining 2.5 µl TaqMan Fast Advance PCR Master Mix (Applied Biosystems, catalog # 4444557), 0.25 µl of 20× GE Sample Loading Reagent (Fluidigm, # PN 100-7610), and 2.5 µl pre-amplified cDNA. IFC controller (Fluidigm, catalog # 68000112 I1) was used to prime the 96.96 dynamic arrays IFC Chip (Fluidigm, catalog # BMK-M-96.96) with the 96.96 control line fluid (Fluidigm, catalog # 89000021). 5 µl of each assay and sample mix was then transferred into the appropriate inlets of the primed chip and loaded with the IFC controller. The PCR was performed at 70°C for 40 min, 60°C for 30 s, 98°C for 1 min, followed by 35 cycles of 97°C for 5 s, 60°C for 20 s. The data was analyzed with Fluidigm Genetic Analysis software. For relative quantification, the amount of target gene expression was normalized to the geometric means of Actb, Gapdh, and Ubb expression, relative to samples from vehicle-dosed controls and was calculated using the $2^{-\Delta\Delta Ct}$ method. The data were analyzed with Real-Time PCR Analysis Software (Fluidigm).

## Assessment of RGC survival

For immunofluorescence analysis of RGC survival and axonal degeneration, the left (ipsilateral to crush) and right eyes were collected 14 days after the ONC procedure and immediately fixed in 4% paraformaldehyde on ice for 1 hr. Retinas were then dissected from the eyes for whole-mount staining. The degree of RGC loss and axonal degeneration at 14 days after ONC was assessed from whole-mount retinas. Floating whole mouse retinas were blocked in 5% normal goat serum in 0.5% Triton X-100 in PBS at 4°C for 2 hr. Primary antibodies were incubated over two nights at 4°C, followed by rinsing and incubation with secondary antibodies in PBS. Immunohistochemistry was performed using the following antibodies: rabbit anti γ-synuclein (1:200, Abcam, ab55424) and mouse anti-TuJ1 (1:40, BioLegend, 801203). Retinas were mounted on slides after being cut with four incisions along the radial axes. Retinas were imaged using a spinning disk confocal microscope (Cell Observer CSU-X1, Carl Zeiss MicroSystem). Images were processed for brightness and contrast in Adobe. RGCs, identified as TuJ1 and γ-synuclein double-immunopositive cells, were quantified manually. Retinal axon degeneration was scored based on density of axon bundles immunolabeled with TUJ-1, using a graded scale from 0 to 5, where 0 corresponds to no degeneration, and 5 to complete degeneration.

## Detection of plasma NfL

Plasma NfL concentrations were measured using Simoa NF-Light Advantage (SR-X version, Quanterix Catalog # 103400) or the Simoa Neurology 4-plex A Advantage (HD-X version, Quanterix Catalog

# 102153) bead-based digital ELISA kit and read on the Quanterix SR-X or HD-X instrument. All experiments were performed according to kit and instrument instructions as provided by Quanterix. All additional equipment and consumables were procured through Quanterix. Plasma samples were diluted 20× with Sample Diluent (Quanterix Catalog # 103399) before being added to Simoa 96-well microplates (Quanterix Catalog # 101457) alongside kit calibrators and controls. Kit-specific instructions were followed and sample NfL levels were measured using the NF Light analysis protocol on the Quanterix SR-X or Neurology 4-Plex A analysis protocol on the Quanterix HD-X instrument and interpolated against a calibration curve provided with the Quanterix assay kit. Mouse NfL data generated from the two instruments have been established to be comparable (data not shown).

## Detection of plasma GFAP
Plasma GFAP was measured using the Simoa Neurology 4-plex A bead-based digital ELISA kit and read on the Quanterix HD-X instrument as described above.

## Preparation of brain homogenate for protein analyses
Brains were homogenized in Radio-Immunoprecipitation Assay Buffer (Teknova) made with cOmplete Protease Inhibitor (Roche, catalog # 04693132001) and PhosStop (Roche, catalog # 04906837001) using the QIAGEN TissueLyzer II (Cat No./ID: 85300) for two rounds of 3 min at 30 Hz. Homogenate was incubated on ice for 20 min and spun at $21,100 \times g$ for 30 min at 4°C. Subsequent lysate was transferred to a clean 96-well deep plate. Protein concentration of the supernatant was measured with the bicinchoninic acid assay (Pierce BCA Protein Assay Kit).

## Detection of GDF-15, TIMP-1, and MCP-1
ELISA kits were used to determine the levels of GDF-15 (mouse/rat GDF-15 Quantikine ELISA Kit, R&D Systems, catalog # MGD150), TIMP-1 (mouse TIMP-1 Quantikine ELISA Kit, R&D Systems, catalog # MTM100), and MCP-1 (Mouse CCL2/JE/MCP-1 Quantikine ELISA Kit, R&D Systems, catalog # MJE00B) in the brain, plasma, or CSF samples.

## RNA-seq library preparation
RNA from bulk mouse brain tissue was extracted using the RNeasy Plus Mini Kit (QIAGEN) and resuspended in nuclease-free water. Libraries were generated using the QuantSeq 3' mRNA-seq Library Prep Kit FWD for Illumina (Lexogen A01173) with the UMI second strand synthesis module to identify and remove PCR duplicates, following the protocol defined by the manufacturer. Briefly, total RNA was used as input for oligo dT priming during reverse transcription, followed by RNA removal. Unique Molecular Identifiers were introduced during second strand synthesis and cDNA was purified using magnetic beads followed by 14 cycles of PCR with dual indexes and PCR purification. Library quantity and quality were assessed with Qubit 1X dsDNA HS Assay Kits (Invitrogen Q33231) and Bioanalyzer High Sense DNA chip (Agilent 5067-4626). Libraries were pooled in equimolar ratios and sequencing reads were generated on an Illumina NovaSeq 6000 instrument (100 bp single end) by SeqMatic (Fremont, CA, USA).

## RNA-seq data preprocessing
Sequencing adapters were trimmed from the raw reads with skewer (version 0.2.2) (*Jiang et al., 2014*). Reads were aligned to the mouse genome version GRCm38_p6. A STAR index (version 2.7.1a) (*Dobin et al., 2013*) and built with the --sjdbOverhang=50 argument. Splice junctions from Gencode gene models (release M17) were provided via the --sjdbGTFfile argument. STAR alignments were generated with the following parameters: `--outFilterType BySJout, --quantMode TranscriptomeSAM, --outFilterIntronMotifs RemoveNoncanonicalUnannotated, --outSAMstrandField intronMotif, --outSAMattributes NH HI AS nM MD XS and --outSAMunmapped Within`. Alignments were obtained with the following parameters: `--readFilesCommand zcat --outFilterType BySJout --outFilterMultimapNmax 20 --alignSJoverhangMin 8 --alignSJDBoverhangMin 1 --outFilterMismatchNmax 999 --outFilterMismatchNoverLmax 0.6 --alignIntronMin 20 --alignIntronMax 1000000 --alignMatesGapMax 1000000 --quantMode GeneCounts --outSAMunmapped Within --outSAMattributes NH HI AS nM MD XS --outSAMstrandField intronMotif --outSAMtype BAM SortedByCoordinate --outBAMcompression 6`. Gene level

counts were obtained using featureCounts from the subread package (version 1.6.2) (*Liao et al., 2014*). Gene symbols and biotype information were extracted from the Gencode GTF file.

## RNA-seq differential expression analysis

Following alignment and expression quantitation, differential expression analysis was performed using the limma/voom framework (*Law et al., 2014*). Only samples with at least 1 million total counts were included in the analysis. To account for remaining differences in library sequence depths, size factors were calculated with the TMM method (*Robinson and Oshlack, 2010*). Genes below the lower limit of quantitation were excluded using the filterByExpr function from the edgeR R package (version 3.42.4) (*Robinson et al., 2010*). A linear model with the following independent variables was fit for each protein-coding gene with the voomWithQualityWeights function: genotype, sex, and age. The comparison of interest was extracted with the eBayes function from the limma R package (version 3.56.2), applying using the robust empirical Bayes procedure of *Phipson et al., 2016*. Genes with FDR<5% were considered significantly differentially expressed.

## Gene set enrichment analysis

To compare the expression changes caused by eIF2B LOF in our study with previous observations, we retrieved the differential expression results published by *Wong et al., 2019*, and performed GSEA with the geneSetTest algorithm from the limma R package. The most significantly upregulated genes reported by Wong et al. (e.g. top 100, 200, 300, 400, or 500 genes ordered by Bayes factor) were highly enriched among genes with large t-statistics in our dataset (p-value $<10^{-20}$). Similar results were obtained with other GSEA algorithms, including camera and fgsea.

## Proteomic analysis of the eIF2B LOF brains and CSF

Proteomics analysis was conducted at BGI Americas (San Jose, CA, USA). For mouse CSF, samples were processed via an in-solution digest. Briefly, 4 µl CSF per sample was reduced with dithiothreitol (DTT) and alkylated with iodoacetamide (IAM). 100 mM HEPES pH 8.5 was added and the samples were digested overnight with Tryp/LysC. The digestion was quenched with 10% TFA, desalted using C18 PhyTips (PhyNexus) and dried by SpeedVac and reconstituted in 100 mM HEPES, pH 8.5. Samples were assigned respective TMT channels in one of four 8-plexes with the final channel being a pool of all the samples to normalize between TMT plexes. Samples were labeled with respective 8-plex TMT channels at RT for 1 hr. Approximately 1% of sample from each channel were mixed, dried down, stage-tipped, and injected into the mass spec for label check then the reaction was quenched with 5% hydroxylamine, mixed in equal amounts, dried down, and cleaned up using C18 PhyTips (PhyNexus). For mouse CSF, due to the low sample amount no fractionation was performed prior to injection on the MS.

Tissue samples were lysed in 5% SDS in triethylammonium bicarbonate, pH 7.1 and assayed for concentrations. 6 µg of each sample was processed using the STrap MS sample prep device according to the manufacturer's protocol (Protifi). More specifically, samples were reduced with DTT and alkylated with IAM; the IAM reaction was then quenched with DTT; the resulting samples were digested in the STrap device with Trypsin/Lys-C overnight. Digested samples were eluted from the STrap according to the manufacturer's protocol. An aliquot of 1 µg was taken from each eluted sample; and the aliquots were pooled to form a composite for TMT bridging. Four aliquots of 5 µg were taken out as four TMT bridges, with each bridge being used for each TMT batch. All samples were dried by SpeedVac. The samples were reconstituted with 50 mM TEAB pH 8.5 and incubated with their assigned TMT channels for 1 hr at RT. For each of the four TMT batches: the same amount of each TMT-labeled sample was aliquoted; and the aliquots were pooled and mixed with 1% formic acid. The four resulting mixtures were loaded onto the nano LC-MS/MS system for Label Check. After the Label Check was passed, all samples were dried by SpeedVac and reconstituted with 2% formic acid. The reconstituted samples of each batch were combined. The resulting three mixtures were further desalted using EVOLUTE EXPRESS ABN according to the manufacturer's protocol (Biotage). The eluted samples were dried by SpeedVac, reconstituted with high-pH reversed-phase buffer A, and loaded onto the offline HPLC for fractionation. Either 8 or 12 fractions were obtained from each sample batch.

All samples were reconstituted with mobile phase A and loaded onto the nano LC-MS/MS system. The same amount of each reconstituted sample was injected for analysis with TMT method on a QE

HF-X or Orbitrap Eclipse MS. Peptide search was performed with Sequest HT and TMT quantification was performed with the reporter ions node in Proteome Discoverer 2.5 (PD, Thermo Fisher). Data was searched against reviewed mouse sequences in Uniprot, full trypsin digest, and 20 ppm precursor, and 0.5 Da fragment tolerance. N-terminal and Lysine TMT6plex and C carbamidomethyl were set as static modifications, oxidized M and N-terminal acetylation were set as dynamic modifications.

## Metabolomic analysis of the eIF2B LOF brains

Sample preparation: 20 mg of snap-frozen brain tissues were bead homogenized using 5 mm stainless steel bead (QIAGEN, catalog # 69989) in 0.4 ml of methanol:water (80:20 vol/vol; LC-MS grade) solution containing 64 stable internal standards. These samples were homogenized with Tissuelyser for 30 s at 25 Hz in 4°C. Crude homogenates were centrifuged at 21,000×$g$ for 20 min at 4°C. Resulting supernatants were collected and transferred to 96 glass well plates (Analytical Sales, PN 27350). Two µl of CSF samples were diluted with 8 µl of 18 mW water and centrifuged at 21,000×$g$ for 10 min at 4°C. To these samples, 100 µl of MS-grade methanol containing 64 stable isotope internal standards was added and vortexed for 5 min at 4°C. Samples were incubated at –20°C for 1 hr and subsequently centrifuged at 21,000×$g$ for 20 min at 4°C. Resulting supernatants were collected and transferred to 96 glass well plates (Analytical Sales, PN 27350).

## Polar metabolomics analysis

Metabolomics analyses were performed using the UHPLC ExionLC system coupled to Sciex QTRAP 6500+ electrospray mass spectrometer in positive and negative ionization modes. For positive ionization mode, 5 µl of sample was injected on a BEH amide 1.7 µm, 2.1×150 mm column (Waters Corporation, Milford, MA, USA) using a flow rate of 0.40 ml/min at 40°C. Mobile phase A consisted of water with 10 mM ammonium formate+0.1% formic acid. Mobile phase B consisted of acetonitrile with 0.1% formic acid. The gradient was programmed as follows: 0.0–1.0 min at 95% B; 1.0–7.0 min to 50% B; 7.0–7.1 min to 95% B; and 7.1–10.0 min at 95% B. For negative ionization mode, Samples N2 dried and reconstituted in 0.1% formic acid. For each analysis, 5 µl of sample was injected on an Imtakt Intrada Organic Acid 3 µm, 2×150 mm (Imtakt USA, Portland, OR, USA), using a flow rate of 0.2 ml/min at 60°C. For negative ionization mode, mobile phase A consisted of acetonitrile/water/formic acid = 10/90/0.1%. Mobile phase B consisted of acetonitrile/100 mM ammonium formate = 10/90%. The gradient was programmed as follows: 0.0–1.0 min at 0% B; 1.0–7.0 min to 100% B; 7.1 at 0% B; and 7.1–10 min at 0% B. Mass spectrometry setting for the positive mode was as follows: curtain gas at 30 psi; collision gas was set at medium, ion spray voltage at 5500 V; temperature at 600°C; ion source Gas 1 at 50 psi; ion source Gas 2 at 60 psi; entrance potential at 10 V; and collision cell exit potential at 12.5 V. For the negative mode, settings were similar to positive mode with the following modifications: ion spray voltage at –4500 V, entrance potential at –10 V, and collision cell exit potential at –15 V. Peaks were identified and integrated using Multiquant 3.02 (Sciex) based on known retention time and mass transitions and normalized to prespecified stable isotope internal standards. Area ratios were exported as all columns report and used for subsequent data analysis.

## Lipidomics analysis

Methanol extracted samples from above were analyzed in electrospray positive and negative ionization modes. For positive ionization mode, mobile phase A consisted of 60:40 acetonitrile/water (vol/vol) with 10 mM ammonium formate+0.1% formic acid; mobile phase B consisted of 90:10 isopropyl alcohol/acetonitrile (vol/vol) with 10 mM ammonium formate+0.1% formic acid. For negative ionization mode, mobile phase composition was same as in positive mode with the exception that 10 mM ammonium acetate and 0.1% acetic acid were used as fortificants. Mass spectrometer settings for positive ionization mode were as follows: curtain gas at 40 psi; collision gas was set at medium; ion spray voltage at 5500 V; temperature at 250°C; ion source Gas 1 at 55 psi; ion source Gas 2 at 60 psi; entrance potential at 10 V; and collision cell exit potential at 12.5 V. Source settings for negative ionization mode were as follows: curtain gas at 30 psi; collision gas was set at medium; ion spray voltage at –4500 V; temperature at 600°C; ion source Gas 1 at 55 psi; ion source Gas 2 at 60 psi; entrance potential at –10 V; and collision cell exit potential at –15.0 V. Lipidomics analysis in positive and negative ionization modes were initiated with 5 µl of sample injection using a BEH C18 1.7 µm, 2.1×100 mm column (Waters) as stationary phase with 0.25 ml/min flow rate and column temperature

set at 55°C. The gradient was programmed as follows: 0.0–8.0 min from 45% B to 99% B, 8.0–9.0 min at 99% B, 9.0–9.1 min to 45% B, and 9.1–10.0 min at 45% B.

## Data pre-processing and analysis for proteomics and metabolomics

### Proteomics data pre-processing

TMT data pre-processing was performed using the MSstatsTMT package (2.4.1). Data were median centered and $\log_2$ transformed. Features with all missing measurement, shared peptides, or proteins with single feature, features with less than three measurements within each run were removed from the analysis. All protein features with ≤1 feature missing were included in the analysis. Nine proteins with one missing value were imputed using kNN function in VIM package (6.2.2).

### Metabolomics and lipidomics data pre-processing

Metabolite/lipid features detected in at least 70% of samples were retained for further analysis. Partially missing values in 44 metabolite/lipid features were imputed using the kNN function in the VIM package. Unwanted variations in data not stemming from relevant biological factors (experimental groups) of interest were detected and removed using the RUV4 function in the ruv (0.9.7.1) package. Top 20th percentile of most abundant metabolite/lipid features with lowest 10th percentile CV from wild-type control mice were selected as negative control features. RUV adjusted data were used for downstream analysis described below.

## Differential expression analysis for proteomics and metabolomics

Unsupervised principal component analysis was performed using FactoMineR (2.8) package. $\log_2$ transformed data were scaled prior to analysis. Univariate analysis of differentially expressed proteins or metabolites were analyzed using limma (3.54.2). Sample specific quality weights were calculated with arrayWeights default method setting. For linear model analysis, a no-intercept model containing the experimental groups of interest (genotype or combination of genotype, treatment, and dose) and covariates (batch) was created. Following linear modeling with lmFit function, contrasts between groups were created using the contrast.fit function to assess group-wise differences. A robust empirical Bayes moderation was performed with robust argument set to TRUE and minimum logFC set to 1.1. FDR was assessed using the Benjamini-Hochberg procedure. Box plots, volcano plots, and heatmaps were generated using the ggplot2 (3.4.2), EnhancedVolcano (1.14), and pheatmap (1.0.12) R packages. Feature overlap and Venn diagram analysis were performed using the VennDiagram (1.7.3) package. GSEA was performed using the clusterProfiler (4.10) R package. Protein ranks were created using moderated logFC with minimum and maximum gene set size set to 3 and 800, respectively. False discovery adjustment when performed used the Benjamini-Yekutieli procedure. All analyses were conducted using R (4.2.1).

## Statistical analysis

Data are reported as means ± SEM or as indicated in figures. Statistical analysis of data was performed in GraphPad Prism version 8 or later. Analysis of data from animal studies was performed using t test or one-way analysis of variance (ANOVA) with multiple comparison, or two-way ANOVA followed by multiple comparison test for data obtained from repeated measures of the same subjects. Analysis of data from human samples was assessed on $\log_2$ fold change data using Welch's t test.

## Acknowledgements

We thank the VWMD patients and their families for generously donating samples, and we extend our thanks to Dr. Raphael Schiffman and the Baylor Scott & White Research Institute for granting us permission to utilize these valuable patient samples in our research. Finally, the authors dedicate this work to the memory of our beloved colleague and friend, Patrick CG Haddick.

# Additional information

## Competing interests

Ernie Yulyaningsih, Jung H Suh, Melania Fanok, Hilda Solanoy, Ryan Takahashi, Anna I Bakardjiev, Isabel Becerra, N Butch Benitez, Chi-Lu Chiu, Sonnet S Davis, William E Dowdle, Timothy Earr, Anthony A Estrada, Audrey Gill, Connie Ha, Patrick CG Haddick, Kirk R Henne, Martin Larhammar, Amy W-S Leung, Romeo Maciuca, Bahram Memarzadeh, Hoang N Nguyen, Alicia A Nugent, Maksim Osipov, Yingqing Ran, Kevin Rebadulla, Elysia Roche, Thomas Sandmann, Jing Wang, Joseph W Lewcock, Kimberly Scearce-Levie, Lesley A Kane, Pascal E Sanchez: employee and shareholder of Denali Therapeutics. Roni Chau: former employee and shareholder of Denali Therapeutics.

## Funding

| Funder | Grant reference number | Author |
| --- | --- | --- |
| Denali Therapeutics | | Ernie Yulyaningsih |
| | | Jung H Suh |
| | | Melania Fanok |
| | | Roni Chau |
| | | Hilda Solanoy |
| | | Ryan Takahashi |
| | | Anna I Bakardjiev |
| | | Isabel Becerra |
| | | N Butch Benitez |
| | | Chi-Lu Chiu |
| | | Sonnet S Davis |
| | | William E Dowdle |
| | | Timothy Earr |
| | | Anthony A Estrada |
| | | Audrey Gill |
| | | Connie Ha |
| | | Patrick CG Haddick |
| | | Kirk R Henne |
| | | Martin Larhammar |
| | | Amy W-S Leung |
| | | Romeo Maciuca |
| | | Bahram Memarzadeh |
| | | Hoang N Nguyen |
| | | Alicia A Nugent |
| | | Maksim Osipov |
| | | Yingqing Ran |
| | | Kevin Rebadulla |
| | | Elysia Roche |
| | | Thomas Sandmann |
| | | Jing Wang |
| | | Joseph W Lewcock |
| | | Kimberly Scearce-Levie |
| | | Lesley A Kane |
| | | Pascal E Sanchez |

The funders had no role in study design, data collection and interpretation, or the decision to submit the work for publication.

## Author contributions

Ernie Yulyaningsih, Conceptualization, Data curation, Formal analysis, Investigation, Visualization, Methodology, Writing – original draft, Writing – review and editing; Jung H Suh, Lesley A Kane, Conceptualization, Data curation, Formal analysis, Investigation, Visualization, Methodology, Writing – original draft, Project administration, Writing – review and editing; Melania Fanok, Data curation, Formal analysis, Investigation, Visualization, Methodology, Writing – original draft, Writing – review and editing; Roni Chau, Patrick CG Haddick, Conceptualization, Data curation, Investigation, Methodology; Hilda Solanoy, Timothy Earr, Hoang N Nguyen, Alicia A Nugent, Jing Wang, Data curation, Investigation, Methodology; Ryan Takahashi, Anna I Bakardjiev, Isabel Becerra, N Butch Benitez, Sonnet S Davis, Amy W-S Leung, Data curation, Formal analysis, Investigation, Methodology; Chi-Lu Chiu, Audrey Gill, Romeo Maciuca, Data curation, Formal analysis, Validation, Investigation, Visualization, Methodology; William E Dowdle, Data curation, Investigation, Methodology, Project administration;

Anthony A Estrada, Conceptualization, Data curation, Investigation, Project administration; Connie Ha, Data curation, Formal analysis, Validation, Investigation, Methodology; Kirk R Henne, Investigation, Methodology, Project administration; Martin Larhammar, Investigation; Bahram Memarzadeh, Yingqing Ran, Kevin Rebadulla, Elysia Roche, Investigation, Methodology; Maksim Osipov, Pascal E Sanchez, Conceptualization, Data curation, Investigation, Methodology, Writing – review and editing; Thomas Sandmann, Data curation, Formal analysis, Validation, Investigation, Methodology, Writing – review and editing; Joseph W Lewcock, Conceptualization, Data curation, Validation, Investigation, Methodology, Writing – original draft, Project administration, Writing – review and editing; Kimberly Scearce-Levie, Conceptualization, Investigation, Methodology, Writing – original draft, Writing – review and editing

**Author ORCIDs**
Jung H Suh ![ORCID] https://orcid.org/0000-0001-8119-326X
Martin Larhammar ![ORCID] https://orcid.org/0000-0002-1547-6760
Hoang N Nguyen ![ORCID] https://orcid.org/0000-0001-6608-7581
Joseph W Lewcock ![ORCID] https://orcid.org/0000-0003-3012-7881
Lesley A Kane ![ORCID] https://orcid.org/0000-0001-9631-3471

**Ethics**
All procedures in animals were performed with adherence to ethical regulations and protocols approved by Denali Therapeutics Institutional Animal Care and Use Committee.

Reviewer #1 (Public Review): https://doi.org/10.7554/eLife.92173.3.sa1
Reviewer #2 (Public Review): https://doi.org/10.7554/eLife.92173.3.sa2
Reviewer #3 (Public Review): https://doi.org/10.7554/eLife.92173.3.sa3
Author response https://doi.org/10.7554/eLife.92173.3.sa4

# Additional files

**Supplementary files**
• Supplementary file 1. Expression of select integrated stress response (ISR)-dependent transcripts at 2 days after optic nerve crush. Data is presented as mean expression level relative to the 0 mg/kg-treated uncrushed retina controls (Group 1). Statistical significance for DNL343 effects in uncrushed retina or the effect of optic nerve crush on non-drug-treated retina was determined by a two-way ANOVA followed by Šídák's multiple comparisons tests against the 0 mg/kg-treated uncrushed retina controls (Group 2 vs Group 1, or Group 3 vs Group 1, respectively). Statistical significance for DNL343 effects in crushed retina was determined by a two-way ANOVA followed by Dunnett's multiple comparisons test against the 0 mg/kg-treated crushed retina controls (Group 4/5/6 vs Group 3). Red font denotes a statistically significant effect (*, p<0.05. **, p<0.01, ***, p<0.001, ****, p<0.0001).

• Supplementary file 2. Expression of select integrated stress response (ISR)-dependent transcripts at 14 days after optic nerve crush. Data is presented as mean expression level relative to the 0 mg/kg-treated uncrushed retina controls (Group 1). Statistical significance for DNL343 effects in uncrushed retina or the effect of optic nerve crush on non-drug-treated retina was determined by a two-way ANOVA followed by Šídák's multiple comparisons tests against the 0 mg/kg-treated uncrushed retina controls (Group 2 vs Group 1, or Group 3 vs Group 1, respectively). Statistical significance for DNL343 effects in crushed retina was determined by a two-way ANOVA followed by Dunnett's multiple comparisons test against the 0 mg/kg-treated crushed retina controls (Group 4/5/6 vs Group 3). Red font denotes a statistically significant effect (*, p<0.05. **, p<0.01, ***, p<0.001, ****, p<0.0001).

• Supplementary file 3. Expression of select integrated stress response (ISR)-dependent transcripts in the brain of eukaryotic initiation factor 2B (eIF2B) homozygous (HOM) mice relative to wild-type and heterozygous controls. Data is presented as mean expression level relative to the wild-type controls (Group 1). Statistical significance for genotype effect was determined by a two-way ANOVA followed by Dunnett's multiple comparisons tests against the wild-type controls (Group 2 vs Group 1, or Group 3 vs Group 1). Red font denotes a statistically significant effect (*, p<0.05. **, p<0.01, ***, p<0.001, ****, p<0.0001).

• Supplementary file 4. Expression of select integrated stress response (ISR)-dependent transcripts in the peripheral blood mononuclear cell (PBMC) of eukaryotic initiation factor 2B (eIF2B) homozygous (HOM) mice relative to wild-type and heterozygous controls. Data is presented as mean expression level relative to the wild-type controls (Group 1). Statistical significance for genotype effect was determined by a two-way ANOVA followed by Dunnett's multiple comparisons tests against the wild-type controls (Group 2 vs Group 1, or Group 3 vs Group 1).

• Supplementary file 5. Expression of select integrated stress response (ISR)-dependent transcripts in the spleen of eukaryotic initiation factor 2B (eIF2B) homozygous (HOM) mice relative to wild-type controls. Data is presented as mean expression level relative to the wild-type controls (Group 1). Statistical significance for genotype effect was determined by a two-way ANOVA followed by Dunnett's multiple comparisons tests against the wild-type controls (Group 2 vs Group 1).

• Supplementary file 6. Expression of select integrated stress response (ISR)-dependent transcripts in the eukaryotic initiation factor 2B (eIF2B) homozygous (HOM) mice following 13 weeks of prophylactic dosing. Data is presented as mean expression level relative to the 0 mg/kg-treated wild-type controls (Group 1). Statistical significance for DNL343 effects in wild-type animals or the effect of the HOM mutation was determined by a two-way ANOVA followed by Šídák's multiple comparisons tests against the 0 mg/kg-treated wild-type controls (Group 2 vs Group 1, or Group 3 vs Group 1, respectively). Statistical significance for DNL343 effects in the eIF2B HOM genotype was determined by a two-way ANOVA followed by Dunnett's multiple comparisons test against the 0 mg/kg-treated eIF2B HOM group (Group 4/5/6/7 vs Group 3). Red font denotes a statistically significant effect (*, $p < 0.05$. **, $p < 0.01$, ***, $p < 0.001$, ****, $p < 0.0001$).

• Supplementary file 7. Vanishing white matter disease (VWMD) patient and healthy control sample information. (A) VWMD patient and (B) healthy control information. VWMD samples were provided by Dr Schiffman at Baylor University. Healthy control samples were purchased or repurposed control samples from another study. Plasma and CSF samples in the healthy control group are from non-matching/different individuals.

• Supplementary file 8. Expression of select integrated stress response (ISR)-dependent transcripts in the eukaryotic initiation factor 2B (eIF2B) homozygous (HOM) mice following 4 weeks of dosing starting at an advanced disease stage. Data is presented as mean expression level relative to the 0 mg/kg-treated wild-type controls (Group 1). Statistical significance for DNL343 effects in wild-type animals or the effect of the HOM mutation was determined by a two-way ANOVA followed by Dunnett's multiple comparisons tests against the 0 mg/kg-treated wild-type controls (Group 2 vs Group 1, or Group 3 vs Group 1, respectively). Statistical significance for DNL343 effects in the eIF2B HOM genotype was determined by a two-way ANOVA followed by Šídák's multiple comparisons test against the 0 mg/kg-treated eIF2B HOM group (Group 4 vs Group 3). Red font denotes a statistically significant effect (*, $p < 0.05$, **, $p < 0.01$, ***, $p < 0.001$, ****, $p < 0.0001$).

• MDAR checklist

## Data availability

RNA-seq data is available from the NCBI GEO repository as Series GSE240150.

The following dataset was generated:

| Author(s) | Year | Dataset title | Dataset URL | Database and Identifier |
|---|---|---|---|---|
| Sandmann T | 2023 | Transcriptional characterization of the Eif2b5[R191H] loss-of-function mouse model | https://www.ncbi.nlm.nih.gov/geo/query/acc.cgi?acc=GSE240150 | NCBI Gene Expression Omnibus, GSE240150 |

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
