## [Editor Report · eLife assessment]

This study presents **solid** evidence to support the effectiveness of the novel eIF2B activator DNL343 in mitigating the integrated stress response (ISR) and reducing neurodegeneration associated with ISR activation in two mouse models. These **important** findings offer promise for the potential use of DNL343 in treating vanishing white matter disease (VWMD), a rare condition resulting from eIF2B loss of function, and in addressing other neurodegenerative disorders characterized by ISR involvement. The study also identified potential VWMD biomarkers, which hold significance for assessing disease progression and evaluating treatment responses.

---

## [Referee Report · Reviewer #1 (Public Review)]

Summary:

In this study, the authors evaluated a novel eIF2B activator, DNL343, in two mouse models representing different integrated stress response (ISR) forms. They first assessed the pharmacokinetics of DNL343, demonstrating its ability to cross the blood-brain barrier and exhibit good bioavailability. In an acute ISR model induced by optic nerve crush (ONC) injury, DNL343 treatment reduced ISR-induced transcriptional changes and neuronal loss, demonstrating neuroprotective effects. Next, the authors generated an eIF2B loss-of-function mice model by knocking in disease-causing Eif2b5 variants. The model presents a chronic ISR and mimics vanishing white matter disease (VWMD). DNL343 treatment from the pre-symptomatic stage improved body weight and motor functions, corrected transcriptional changes, and reversed proteomic and metabolomic alterations in the brain and cerebrospinal fluid. DNL343 treatment initiated at an advanced disease stage also showed positive effects, restoring body weight gain, suppressing ISR, reducing neurodegeneration biomarkers, and extending lifespan. These findings highlight DNL343 as an effective ISR inhibitor with potential applications in treating VWMD and other neurodegenerative disorders involving ISR.

Strengths:

The study's findings regarding the novel compound DNL343 offer significant promise in addressing VWMD, a condition currently lacking disease-modifying treatment. DNL343 directly targets eIF2B, the disease-causing complex in VWMD, and demonstrates notable efficacy in reversing the integrated stress response (ISR) and mitigating neurodegeneration in a VWMD mouse model. These results raise hope for the potential application of DNL343 in VWMD treatment, a development eagerly anticipated by patients and the VWMD research community. Moreover, the study hints at the broader potential of DNL343 in treating other ISR-related neurodegenerative disorders, such as ALS, a prospect that holds broader interest. Additionally, the study's identification of potential biomarkers for VWMD represents a notable strength, potentially leading to improved disease progression assessment pending further confirmation in future research.

Weaknesses:

Direct biochemical evidence confirming DNL343's activity in eIF2B activation and its toxicity profile have been previously documented in a separate study. It would be beneficial to provide a more detailed introduction to this information, establishing a robust knowledge foundation for the in vivo study described in this work.

---

## [Referee Report · Reviewer #2 (Public Review)]

Summary:

The authors developed DNL343, a CNS-penetrant small molecule integrated stress response (ISR) inhibitor, to treat neurodegenerative diseases caused by ISR.

Strengths:

DNL343 is an investigational CNS-penetrant small molecule integrated stress response (ISR) inhibitor designed to activate the eukaryotic initiation factor 2B (eIF2B) and suppress aberrant ISR activation. The therapeutic efficacy of DNL343 has been extensively characterized in two animal models. Importantly, plasma biomarkers of neuroinflammation and neurodegeneration can be reversed with DNL343 treatment. Remarkably, several of these biomarkers show differential levels in CSF and plasma from patients with vanishing white matter disease (VWMD) upon DNL343 treatment. Overall, this study is very exciting that targets ISR for therapeutic interventions.

Weaknesses:

My main questions center around the characterization of DNL343.

(1) Is there any biochemical evidence showing DNL343 activates eIF2B, such as binding and in vitro biochemical activity assays? A conference presentation was cited. "Osipov, M. (2022). Discovery of DNL343: a Potent Selective and Brain-penetrant eIF2B Activator Designed for the Treatment of Neurodegenerative Diseases. Medicinal Chemistry Gordon Research Conference. New London, NH." However, there is no public information about this presentation.

(2) How was the selectivity of DNL343 demonstrated? What are the off-targets of DNL343, particularly when DNL343 is administered at a high dose? Thermal-proteasome profiling or photoaffinity labeling experiments could be considered.

(3) What are the total drug concentrations in the brain and plasma? What are the unbound ratios?

(4) If DNL343 is given intravenously, what are the concentrations in the brain and plasma after 5 minutes and 1 h or longer time points? In other words, does DNL343 cross BBB through passive diffusion or an active process?

(5) What is the full PK profile of DNL343 for intravenous and oral dosing?

(6) Are there any major drug metabolites that could be concerns?

Review for Revision:

The companion JMC paper, doi.org/10.1021/acs.jmedchem.3c02422, addressed most of my questions. However, I was unable to find the total concentrations of DNL343 in the brain and plasma or the raw data for the full PK in the JMC paper. Otherwise, the JMC publication addressed all my questions.

---

## [Referee Report · Reviewer #3 (Public Review)]

Summary:

ISR contributes to the pathogenesis of multiple neurodegenerative diseases, such as ALS, FTD, VWMD, etc. Targeting ISR is a promising avenue for therapeutic intervention. However, all previously identified ways to target ISR have problems. PERK inhibitors suppress ISR by inhibiting eIF2alpha phosphorylation and cause pancreatic toxicity in mice. In order to bypass eIF2alpha, previous studies have identified ISR suppressors that target eIF2B, such as ISRIB and 2BAct. These molecules suppress neurodegeneration but do not cause detrimental effects in mouse models. However, ISRIB is water-insoluble, and 2BAct causes cardiovascular complications in dogs, preventing their use in clinics. Here, the authors showed that DNL343, a new ISR inhibitor targeting eIF2B, suppresses features that can be related to neurodegeneration in mouse models. Combined with their previous results of a clinical phase I trial showing the safety of DNL343, these findings suggest the promise of DNL343 as a potential drug for neurodegenerative diseases in which ISR contributes to pathogenesis.

Strengths:

The finding is important and has disease implications.

Weakness:

The authors did not provide evidence that DNL343 suppresses the demise of nervous systems in their VWMD model.

---

## [Author Response]

The following is the authors’ response to the original reviews.

**Public Reviews:**

**Reviewer #1 (Public Review):**
Summary:In this study, the authors evaluated a novel eIF2B activator, DNL343, in two mouse models representing different forms of the integrated stress response (ISR). They first assessed the pharmacokinetics of DNL343, demonstrating its ability to cross the blood-brain barrier and exhibit good bioavailability. In an acute ISR model induced by optic nerve crush (ONC) injury, DNL343 treatment reduced ISR-induced transcriptional changes and neuronal loss, demonstrating neuroprotective effects. Next, the authors generated an eIF2B loss-of-function mice model by knocking in disease-causing Eif2b5 variants. The model presents a chronic ISR and mimics vanishing white matter disease (VWMD). DNL343 treatment from the pre-symptomatic stage improved body weight and motor functions corrected transcriptional changes, and reversed proteomic and metabolomic alterations in the brain and cerebrospinal fluid. DNL343 treatment initiated at an advanced disease stage also showed positive effects, restoring body weight gain, suppressing ISR, reducing neurodegeneration biomarkers, and extending lifespan. These findings highlight DNL343 as an effective ISR inhibitor with potential applications in treating VWMD and other neurodegenerative disorders involving ISR.Strengths:The study's findings regarding the novel compound DNL343 offer significant promise in addressing VWMD, a condition currently lacking disease-modifying treatment. DNL343 directly targets eIF2B, the disease-causing complex in VWMD, and demonstrates notable efficacy in reversing the integrated stress response (ISR) and mitigating neurodegeneration in a VWMD mouse model. These results raise hope for the potential application of DNL343 in VWMD treatment, a development eagerly anticipated by patients and the VWMD research community. Moreover, the study hints at the broader potential of DNL343 in treating other ISR-related neurodegenerative disorders, such as amyotrophic lateral sclerosis, a prospect that holds broader interest. Additionally, the study's identification of potential biomarkers for VWMD represents a notable strength, potentially leading to improved disease progression assessment pending further confirmation in future research.Weaknesses:There are a couple of notable concerns in this study. Firstly, while the in vivo evidence strongly supports the efficacy of DNL343 in mitigating ISR and neurodegeneration, there is a lack of direct biochemical evidence to confirm its activity in eIF2B activation. Secondly, the potential for cardiovascular toxicity, which has been reported for a related eIF2B activator in a canine model (as mentioned in the manuscript), has not been evaluated for DNL343 in this study. This data gap regarding toxicity could be crucial for informing the future development of DNL343 for potential human use. Further investigation into these areas would be valuable for a comprehensive understanding of the compound's mechanisms and safety profile.

We thank the reviewer for the thoughtful feedback and an opportunity to provide further clarification. To address the first question regarding biochemical evidence of the mechanism of action of DNL343, we agree that additional data is helpful to interpreting the results presented in this manuscript. We now include a citation to Craig et al (Craig, R.A., 2nd, J. De Vicente, A.A. Estrada, J.A. Feng, K.W. Lexa, M.J. Canet, W.E. Dowdle, R.I. Erickson, B.N. Flores, P.C.G. Haddick, L.A. Kane, J.W. Lewcock, N.J. Moerke, S.B. Poda, Z. Sweeney, R.H. Takahashi, V. Tong, J. Wang, E. Yulyaningsih, H. Solanoy, K. Scearce-Levie, P.E. Sanchez, L. Tang, M. Xu, R. Zhang and M. Osipov (2024). "Discovery of DNL343: A Potent, Selective, and Brain-Penetrant eIF2B Activator Designed for the Treatment of Neurodegenerative Diseases." *J Med Chem*.) which includes the full details on the discovery and characterization of DNL343.

On the question of cardiovascular toxicity observed with previous eIF2B activating compounds, Craig et al also provides evidence in a non-human primate (cynomolgus monkey) model that DNL343 dosing did not result in QT prolongation or any functional cardiac changes. We have also completed a Phase 1 (NCT04268784) and Phase 1B double-blind (NCT05006352) trials in healthy and ALS participants, respectively and these trials are referenced on page 4, lines 102-103. The safety profile observed in these clinical studies supported further development of DNL343 for ALS in the Healey Platform trial (NCT04297683, Regimen G).

**Reviewer #2 (Public Review):**
Summary:The authors developed DNL343, a CNS-penetrant small molecule integrated stress response (ISR) inhibitor, to treat neurodegenerative diseases caused by ISR.Strengths:DNL343 is an investigational CNS-penetrant small molecule integrated stress response (ISR) inhibitor designed to activate the eukaryotic initiation factor 2B (eIF2B) and suppress aberrant ISR activation. The therapeutic efficacy of DNL343 has been extensively characterized in two animal models. Importantly, plasma biomarkers of neuroinflammation and neurodegeneration can be reversed with DNL343 treatment. Remarkably, several of these biomarkers show differential levels in CSF and plasma from patients with vanishing white matter disease (VWMD) upon DNL343 treatment. Overall, this is a very exciting study to target ISR for therapeutic interventions.Weaknesses:My main questions center around the characterization of DNL343.(1) Is there any biochemical evidence showing DNL343 activates eIF2B, such as binding assays or in vitro biochemical activity assays? A conference presentation was cited - "Osipov, M. (2022). Discovery of DNL343: a Potent Selective and Brain-penetrant eIF2B Activator Designed for the Treatment of Neurodegenerative Diseases. Medicinal Chemistry Gordon Research Conference. New London, NH." However, there needs to be public information about this presentation.

Information from this presentation and more details on the discovery and characterization of DNL343 can be found in Craig et al *J Med Chem* (2024) and this citation has been replaced.

(2) How was the selectivity of DNL343 demonstrated? What are the off-targets of DNL343, in particular when DNL343 is administered at a high dose? Thermal-proteasome profiling or photoaffinity labeling experiments could be considered.

Please see Craig et al *J Med Chem* (2024) for full details. In brief, there were no significant off target effects observed for DNL343 in a Cerep panel.

(3) What are the total drug concentrations in the brain and plasma? What are the unbound ratios?

Following a single oral dose of DNL343 in mice, unbound brain-to-unbound plasma exposures ratios (Kp,uu) of 0.8 to 1.1 were observed, indicating high CNS penetrance. This was further supported by CSF-to-unbound plasma exposures ratios at 0.9 in the same mouse study. The CNS penetrance was also confirmed in rats and NHP by CSF-to-unbound plasma ratios near unity as reported in Craig et al *J Med Chem* (2024).

(4) If DNL343 is given intravenously, what are the concentrations in the brain and plasma after 5 minutes and 1 hour or longer time points? In other words, does DNL343 cross BBB through passive diffusion or an active process?

Unbound brain-to-unbound plasma exposure ratios following a single oral dose in the mouse were 0.8 to 1.1 and showed no time dependence. These measurements were made prior to, near, and following plasma tmax of DNL343, indicating unbound DNL343 crosses the BBB through passive diffusion and rapidly reached equilibrium between the brain and systemic circulation. Details can be found in Craig et al *J Med Chem* (2024).

(5) What is the complete PK profile of DNL343 for intravenous and oral dosing?

DNL343 administered orally to mice as a suspension formulation showed plasma PK consistent with prolonged absorption with tmax ranging from 3 to 4 h, and a terminal elimination half-life (t1/2) of ~10 h. Details can be found in Craig et al *J Med Chem* (2024).

(6) Are there any major drug metabolites that could be of concern?

DNL343 metabolism is through Phase 1 biotransformation pathways. None of the in vivo circulating metabolites show potency towards eIF2B activation. Given that none of these metabolites are of concern, we believe this information is beyond the scope of the current manuscript.

**Reviewer #3 (Public Review):**
Summary:ISR contributes to the pathogenesis of multiple neurodegenerative diseases, such as ALS, FTD, VWMD, etc. Targeting ISR is a promising avenue for potential therapeutics. However, previously identified ways to target ISR present some challenges. PERK inhibitors suppress ISR by inhibiting eIF2alpha phosphorylation and cause pancreatic toxicity in mice. In order to bypass eIF2alpha, previous studies have identified ISR suppressors that target eIF2B, such as ISRIB and 2BAct. These molecules suppress neurodegeneration but do not cause detrimental effects in mouse models. However, ISRIB is water-insoluble, and 2BAct causes cardiovascular complications in dogs, preventing their use in clinics. Here, the authors showed that DNL343, a new ISR inhibitor targeting eIF2B, suppresses neurodegeneration in mouse models. Combined with their previous results of a clinical phase I trial showing the safety of DNL343, these findings suggest the promise of DNL343 as a potential drug for neurodegenerative diseases in which ISR contributes to pathogenesis.Strengths:The finding is important and has disease implications, and the conclusion is not surprising.Weaknesses:The experimental design and data are hard to comprehend for an audience with a basic research background. This reviewer suggests that the authors use the same way that previous studies on ISRIB and 2BAct (e.g., Wong et al; eLife, 2019) designed experiments and interpret data.

We thank this reviewer for their feedback and recognition that DNL343 has a promising potential as treatment for neurodegenerative diseases. While our studies share some similarities to Wong et al., *eLife* (2019) and Abbink et al., *ACTN* (2019), our study design is intentionally distinct (e.g. inclusion of both prevention and treatment dosing paradigms, determining dose-response impact of drug treatment across biomarkers) which necessitates tailored data visualization to effectively communicate our findings. However, we understand the importance of clarity for a broader audience and to this end, we have made a number of changes to the data figures, in particular data from omics experiments in Figures 3 and 5. We also provided additional supplemental tables to aid data interpretation. This would hopefully cater to both audiences familiar with previous work and those with a less specialized background.

**Recommendations for the authors:**

**Reviewer #1 (Recommendations For The Authors):**
(1) Demyelination is a significant pathological feature in the VWMD mouse model. The authors should clarify whether they observed similar demyelination in their study and if DNL343 had any impact on reversing this demyelination. These findings are crucial for assessing the compound's effectiveness in mitigating neurodegeneration.

Demyelination is indeed an important feature in the eIF2B LOF (VWMD) mouse model. Given that this phenotype and the ability to rescue the histological phenotype with this MOA (Wong et al; eLife, 2019, cited in introduction) is very well characterized, along with our limitation from the size and number of mouse tissues, we prioritized non-histological targeted and unbiased analyses that were aimed at identifying translatable biomarkers. Nonetheless, the totality of our data, in different mouse models and cell types, strongly supports DNL343 as a potent ISR inhibitor that is effective in attenuating neurodegeneration:

· In the optic nerve crush model, DNL343 dose-dependently reduced retinal cell degeneration

· In the VWMD mouse model, DNL343 attenuated the increase in a plasma biomarker of neurodegeneration, neurofilament-light, which corresponded to normalization in motor function.

· Metabolomic and lipidomic analyses in the VWMD mouse model brain showed increases in oxysterols, such as 7-ketocholesterol, and cholesterol esters and these lipids are associated with demyelination (Nugent et al, 2020). DNL343 treatment attenuated the levels of these oxysterols, indicating decreased demyelination.

· When initiated at an advance disease stage, reversal of plasma biomarkers of neurodegeneration (Nf-L) and neuroinflammation (GFAP) by DNL343 in this model was accompanied by extension in the lifespan that is otherwise shortened as the mutant animals succumb to disease.

These data highlight the potential therapeutic benefits of DNL343 in the broader context of ISR-mediated neurodegeneration which can include but may not be limited to VWMD.

(2) Figure 6 presents several biomarkers with significantly increased levels in VWMD mice and patient biofluids. However, these biomarkers are not reflected in the brain proteomics data presented in Figure 3. The discrepancy between these findings should be addressed and discussed in the manuscript to provide a more comprehensive understanding.

Proteins detected in Figure 6 were not detected by TMT proteomics in the CSF. In the brain, only GFAP was detected and the overall abundance in tissue were similar in both genetic groups. Cytokines such as TIMP1, MCP1 are usually present in low abundances and therefore are challenging to detect in broad discovery proteomics method applied in this study. Antibody-based immunoassays are better suited to specifically measure low abundant proteins than mass-spectrometry-based proteomics, while mass-spectrometry based methods offer wider dynamic range to detect more highly abundant proteins. Differences in detection sensitivity between immunoassay vs mass spectrometry assays has been previously noted (Petrera et al, J Proteome Res, 2021). We have added new text to address this point in the revised manuscript (page 7, line 274-277).

(3) Figure 7 discusses the effects of DNL343 treatment initiated at an advanced disease stage. Since the 4-week treatment did not rescue performance in the balance beam test (as shown in Figure 6A), it is important to clarify if a 20-week treatment had any impact on this parameter.

This reviewer raised an important question that we were unfortunately unable test. When the balance beam training was administered after 8 (out of 20) weeks of dosing, most animals of both wildtype and mutant genotypes struggled to remain on or maintain balance on the beam and were unable to progress traversing the beam, making the assay unsuccessful in this cohort. This impairment appeared to be driven by distinct factors in the two genotypes: age-associated obesity in wild-type animals and severe motor impairment in the eIF2B HOM mice, irrespective of treatment. While it is possible that other less demanding and more sensitive assays could reveal more nuanced differences, this, and our earlier data (Figure 4G-I), suggest that DNL343 could prevent but not reverse functional deterioration. This is in line with our understanding of DNL343 mechanism of action that does not include neuronal regeneration, a therapeutic effect that is likely required for functional recuperation. We have added this point to the manuscript (page 8, line 319-326).

Additionally, considering the significant increase in Gdf15 levels in the disease model, it would be valuable to know if DNL343 treatment affected Gdf15 levels. If these assays were conducted, reporting the data would greatly assist in evaluating the compound's efficacy when administered at an advanced disease stage.

We were not able to measure GDF15 levels in the 20-week study due to limitation in the in-life collected plasma samples which was dedicated to assessing biomarkers of neurodegeneration (Figure 7E-F). However, data from our 4-week treatment study, which was initiated at a similar age range to the 20-week treatment study (19-26 and 24-33 weeks of age, respectively), showed that DNL343 was able to reduce GDF15 levels in the brain (mRNA and protein) and CSF (protein) (Supplemental Figure 5A-C), suggesting that DNL343 reduces ISR activation at an advanced disease stage in the model. We expect that this reduction observed at 4 weeks of treatment would persist for the duration of the extended treatment in the 20-week cohort.

(4) A minor point. In Figures 5A, 5C, and 5E, it appears that the red-colored group should likely be labeled as "HOM 0 mg/kg" instead of "HOM 3 mg/kg".

This has been amended, thank you.

**Reviewer #3 (Recommendations For The Authors):**
Major concerns:(1) The cellular function of DNL343 needs to be clarified. The authors claim that it activates eIF2B, but no cellular or molecular evidence is provided. Does it bind to eIF2B? Does it not affect eIF2alpha phosphorylation? Does it restore translation upon stress that causes eIF2alpha phosphorylation? Does it suppress stress granule assembly? The authors cited Sun, Tsai et al. 2023 and Osipov et al., 2022. However, these citations are conference abstracts with no published figures available for review.

We agree that additional data outlining the biochemical evidence of the mechanism of action of DNL343 was needed. We now include a citation to Craig et al *J Med Chem* (2024) that includes the full details on the discovery and molecular characterization of DNL343.

(2) It needs to be clarified how the authors selected the ISR marker genes. ISR genes are more than those selected. How about others? How did the authors measure the mRNA levels, bulk RNA-seq or RT-PCR? If the former, have the authors verified their results using RT-PCR? Have the authors measured the protein levels for nerve crush experiments (by both proteomic and individual protein analyses)? Also, no statistical analyses were found for the heat maps.

The ISR marker genes were selected by a combination of experimental and literature data. Transcriptomics analysis of the eIF2B HOM brains was conducted using untargeted RNAseq (Supplemental Figure 1B). Here, we found an enrichment of transcripts previously reported to be ISR dependent, namely *Atf4*, *Chac1*, *Ddit3*, *Eif4ebp1*, *Ppp1r15a* (Larhammar et al., 2017), *Atf3*, *Asns*, *Mthfd2*, *Psat1*, *Sesn2*, *Slc1a5*, *Slc7a5*, *Slc7a11*, *Trib3* (Wong et al., 2019, Abbink et al., 2019). These transcripts were assayed using targeted qPCR in the eIF2B HOM brains, spleen and PBMC (Supplemental Figure 1A, C, D) and in the retinas from the ONC experiments (Figure 2C). We have further clarified the analysis method for the gene expression data in the figure legends.

We did not interrogate the proteome of the retina in the ONC model. Our study in this model was intended as a proof-of-concept evaluation of DNL343 effects in this acute ISR-dependent model of neurodegeneration. To this end, we performed gene expression (Figure 2C) and immunofluorescence analyses (Figure 2D-F). Each of these analyses were conducted using dedicated whole retinas; conducting additional protein analyses would necessitate a separate cohort of animals.

We believe that heatmaps provide the best visualization of the data, particularly the dose dependent effects of DNL343 on multiple genes, but we understand the value for also providing statistical analyses. To address this, we provide additional Supplemental tables to show the outcome of statistical analyses undertaken. Statistical data relating to Figure 2C can be found on new Supplemental Tables 1 & 2; those relating to Supplemental Figures 1A, C, and D on new Supplemental Tables 3, 5, 6, respectively; that from Figure 4D on new Supplemental Table 8, and that from Figure 7D on new Supplemental Table 11.

(3) Both the authors and Wong et al. (eLife, 2019) performed transcriptomic analyses on HOM mice. How do the authors compare the two data sets? Are they the same?

In this work, transcriptomic approach was applied to confirm induction of ISR response in our in vivo model. While data are not identical, all of the top annotated genes shown in supplementary figure 1B were also deemed to be significant by Wong and coworkers (Bayes factor > 10). More importantly, as explained in our responses to question #2 from reviewer 3, ISR genes highlighted in supplementary Figure 1B were also confirmed in two other studies (Larhammar et al., 2017, Abbink et al., 2019). These data support our interpretation that eIF2B HOM have elevated ISR relative to WT mice. We have added new text to line 164 on page 5 to clarify this point.

(4) Can the authors interpret their omic data using volcano plots for HOM rescue experiments, as Wong et al. did in eLife 2019? Heat maps with statistical analyses are more straightforward to comprehend. Can the authors verify some of these data using RT-PCR, Western blot, etc.?

We added additional pathway interpretation in our Figure 3 and 5 to highlight key biological processes altered in the brain and cellular compartment origin of CSF proteins changed in eIF2B HOM at baseline and following treatment with DNL343. Our treatment designed employed multiple dosing levels and as such, summarization by volcano plot would have resulted in creation of many figures that can be more easily captured by a single heat map plot. However, to provide additional quantitative information, we now added supplementary tables showing full statistical analysis for all heat maps for added clarity and transparency.

We demonstrated 100% correlation between the select genes we examined by qPCR in supplemental Figure 1A and those identified from brain by RNA-seq. In addition, question of reliability of RNA-seq data has been previously been examined in great detail (Everaet et al, *Sci Rep* 2017) and found ~85% concordance between RNA-seq and qPCR data and those that were discordant tended to have < 2 log2FC and were present in low abundance. Given that top core ISR genes identified in our study have >2 log2FC and have been verified by other independent labs (Larhammar et al., 2017, Abbink et al., 2019, Wong et al., 2019). Based on these, we do not think that there is a rationale need for technical confirmation of RNAseq data.

Risks for mis-annotation of proteins in TMT data were further mitigated by removing protein with coverage < 20% and having less than 8 unique peptides detected and setting protein annotation FDR to <1%.

Additionally, TMT-labelling based proteomics offers wider dynamic range and sensitivity than western blotting. Validation of TMT logFC data with western blot technique, which is less quantitative and has lower dynamic ranges of detection may not be very informative. Furthermore, similar trends of changes in key ISR genes and proteins shown in figures 4D and 5A (e.g PSAT, SLC7A11, SLC7A5) provides additional support for the authenticity of proteins identified in this work.

Also, for Figures 4E and F, it is assumed that each line represents an individual animal, but why their body weight gains are so different for the wild type? Can the authors plot the mean and s.e.m.? Also, there are no data about neurodegeneration. The authors need to show microscopy images, count the numbers, and assess the morphology of nerve cells.

The large data spread in the body weight gain in our wild-type mice reflect the normal variability of this endpoint which can be influenced by sex and age. Indeed, both factors are present in our cohorts as animals of both sexes were included and there was a 7-week age-range (10-17 weeks of age at dosing start). Each line in Figures 4E-F indeed represents data sampled from individual animal over time. We chose to represent the data this way for transparency and have provided additional visualization (new Supplemental Figure 3) showing both body weight gain and plasma Nf-L levels as mean ± SEM as requested by this reviewer.

In this study we chose to use a clinically-relevant biomarker of neurodegeneration, plasma neurofilament light chain (NfL) (Figure 4F). This allowed us to prioritize the tissue samples from these studies to execute comprehensive unbiased analyses for more complete characterization of the phenotype of these eIF2B LoF mice. NfL is a biomarker that has been recognized as a sensitive measurement of neuronal/axonal damage regardless of cause (Gaetani et al., 2018, Khalil et al., 2018). Elevated levels of plasma (and CSF) NfL levels has been demonstrated across neurodegenerative conditions such as Alzheimer’s disease (Giacomucci et al., 2022), multiple sclerosis (Ferreira-Atuesta et al., 2021), and in ALS (Huang et al., 2018).

(5) How ISR is connected to metabolomic changes? Can the authors explain it?

ISR caused significant increases in amino acid transporter and serine/glycine/1-carbon metabolism enzymes transcript and protein abundances that were highlighted in Figure 3A and C and lines 237-255 in the main text. Similar patterns were also observed in prior published studies (Larhammar et al., 2017, Abbink et al., 2019, Wong et al., 2019). Consistent with these changes we observed increased levels of Alanine (transported by SLC3A2, SLC7A11, SLC7A3) and decreased cystathionine levels (associated with increased expression of CTH). ATF4 is one of the main orchestrator of ISR response to stress (e.g., amino acid deprivation) and it is required for expression of amino acid transporters and enzymes required for synthesis non-essential amino acids (PMID: 28494858). ATF4 increases cellular amino acid uptake and deliver AA needed for synthesis of proteins and glutathione needed for survival.

We also observed prominent changes in CE in eIF2B HOM and its normalization with DNL343 treatment shown in Figure 5C. We checked for changes in expression levels of CEL, CES1, LCAT, LIPA, SOAT1, and NCEH1 proteins involved in CE metabolism and failed to detect any changes in protein or RNA abundances. This suggests that a rapid demyelination is a more likely trigger for CE accumulation as reported in FTD-GRN (Marian OC et al., 2023 *acta neuropathol commun*
**11**, 52), and in experimental demyelination models (Nugent AA et al., 2020 Neuron). We have added new text to the discussion section of the manuscript page 9, lines 408-411 to discuss how these results relate to each other.

(6) It is hard to understand the biomarker part. The authors said "potential translational biomarkers are elevated..." Do the authors mean they are elevated so they can be potential biomarkers? If their levels are unchanged (e.g., TIMP-1), how can they be biomarkers? Also, this part needs a conclusion/summary. Also, what does "reversed biomarkers..." mean?

We have modified the text to clarify and included a concluding sentence for this section of the results (page 7, lines 297-299). In assessing whether a given protein could be a potential translational biomarker for human disease we evaluated if the following two conditions were met: (1) Increased or decreased gene expression or protein levels of the biomarker in the brain or biofluids (CSF or plasma) of *Eif2b5 R191H* homozygote mice relative to wild-type controls that is modulated or normalized by administration of DNL343 and (2) protein levels in biofluids from VWMD patients that show differential levels than healthy controls in the same directionality as what is seen in the mouse model. GDF-15, GFAP, and NfL meet these criteria, but TIMP-1 and MCP-1 do not.

Minor concerns:(1) Please explain which multiple comparison tests the authors used.

This information has been further clarified in the figure legends.

(2) Administrating the drug at an advanced stage led to a trend of NfL reduction but did not rescue function. Can the authors discuss what this means?

Further elaboration and discussion about this finding have been added to the results section on page 8, line 319-325.

(3) For statistical analyses on the bar graphs, it would be better if the authors labeled the comparison pairs on the graphs.

We agree that labelling comparisons in bar graphs could aid the readership and have added this modification. Additionally, comparisons are indicated in the figure legend.

(4) The authors need to state clearly that 2BAct's cardiovascular toxicity was observed in dogs, not mice. The current study does not exclude similar DNL343 toxicity. However, previous clinical trials suggest that DNL343 may be safe for humans.

The suggestion to specify cardiovascular toxicity in dogs has been added (page 3, line 101), thank you. We now include a citation to Craig et al *J Med Chem* (2024) that provides evidence in a non-human primate (cynomolgus monkey) model that DNL343 dosing did not result in QT prolongation or any functional cardiac changes. We have also completed a Phase 1 (NCT04268784) and Phase 1B double-blind (NCT05006352) trials in healthy and ALS participants, respectively and now include reference to these trials on page 4, lines 102-104. The safety profile observed in these clinical studies supported further development of DNL343 for ALS in the Healey Platform trial (NCT04297683, Regimen G).